# FedPDG: Prediction Discrepancy–Guided Data Generation for Heterogeneous Federated Learning

Yuqi Wang [* 1]   Jianwei Niu [1 2 3]   Xinghao Wu [* 1]   Xuefeng Liu [1 2 †]   Xin Hao [1]

## Abstract

One emerging approach to mitigating data heterogeneity in Federated Learning (FL) is to employ diffusion models to generate synthetic data for clients, thereby aligning local data distributions with the global distribution. Prior work has primarily focused on balance-oriented augmentation, which assumes a balanced global class distribution and thus generates samples of rare classes to rebalance each client's local dataset. However, in practice, global data distributions are often inherently imbalanced. Moreover, privacy constraints in FL hinder the server's ability to accurately estimate the global distribution, rendering balance-oriented augmentation suboptimal. This raises a key, underexplored challenge: How can synthetic data be generated and selected to align local distributions with the true, yet unknown, global distribution? Our key insight is that a model's performance implicitly reflects the data distribution it has been trained on. Based on this observation, we use the performance discrepancy between local and global models to identify the regions where each client's local dataset is lacking, and generate corresponding samples for clients. Furthermore, we adapt the diffusion model via preference optimization, enabling it to generate data that better aligns with the true global distribution. Extensive experiments on multiple benchmarks demonstrate that FedPDG outperforms state-of-the-art methods, achieving up to 3.82% improvement.

---

[*]Equal contribution  [1]State Key Laboratory of Virtual Reality Technology and Systems, School of Software, Beihang University, Beijing, China [2]Hangzhou Innovation Institute of Beihang University, Zhejiang Key Laboratory of Industrial Big Data and Robot Intelligent Systems, Hangzhou, China [3]Center for AI Business Innovation, Department of Management Science and Systems, University at Buffalo, Buffalo, New York, USA. Correspondence to: Xuefeng Liu <liu_xuefeng@buaa.edu.cn>.

## 1. Introduction

Federated Learning (FL) (McMahan et al., 2017) enables decentralized clients to collaboratively train models without sharing raw data. A key challenge in FL is data heterogeneity, where clients possess non-Independent and Identically Distributed (non-IID) data. This heterogeneity often leads to model drift, where local models diverge from the global model during training, amplifying discrepancies between clients and ultimately degrading the global model's performance. To mitigate this drift, existing methods constrain local updates or align local models with the global objective (Sahu et al., 2018; Karimireddy et al., 2019; Li et al., 2021; Acar et al., 2021; Luo et al., 2021; Zhang et al., 2022). However, these approaches introduce a trade-off between fitting local data and maintaining alignment with the global model, often compromising the ability of local models to learn client-specific knowledge, especially when data heterogeneity is high.

To tackle these challenges, recent efforts have explored the use of diffusion models to generate synthetic data for clients, aiming to shift their local data distributions to align more closely with the global distribution, thereby reducing model drift during local training. Compared to regularization-based methods, this paradigm mitigates data heterogeneity at the source, thus avoiding introducing the trade-off between fitting local data and aligning with the global model.

We observe that existing methods for data supplementation in Federated Learning (FL), such as (Wen et al., 2022; Morafah et al., 2024; Qiang et al., 2025), commonly assume a *balanced global label distribution*, and accordingly generate synthetic samples to balance the class distribution within each client's local dataset. This assumption is reasonable in many critical domains, such as healthcare and autonomous driving, where the goal is to mitigate model bias and reduce false positives. In these scenarios, although individual clients may have imbalanced data, it is desirable for the global training and testing distributions to be balanced, so as to ensure fairness and minimize the risk of costly mispredictions.

However, in many real-world applications, *the global data distribution is inherently imbalanced*, and such imbalance is

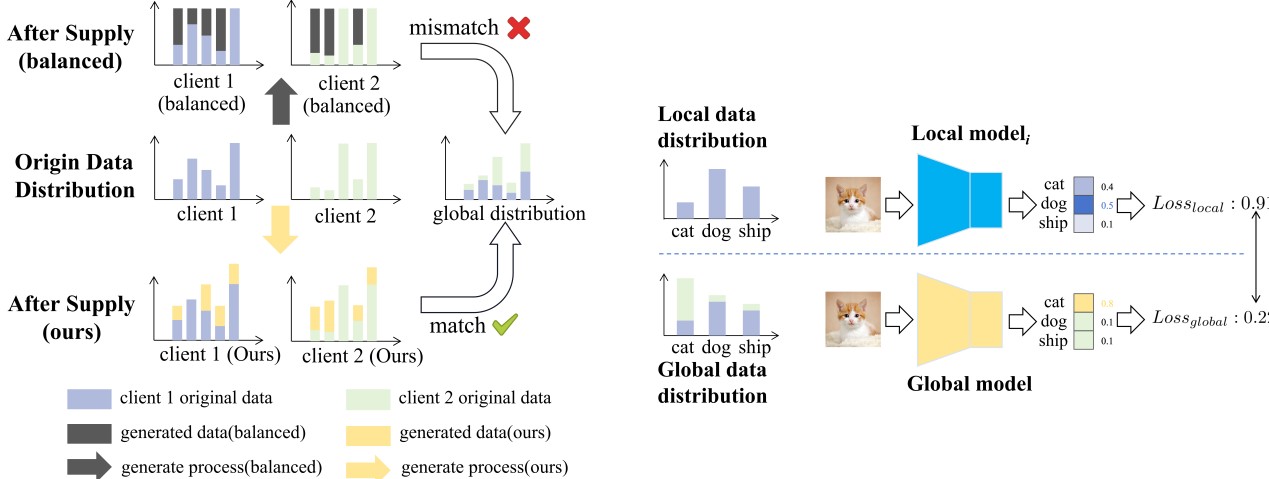

*(a)* Comparing balanced supplement and ours supplement.

*(b)* Illustration of our motivation leveraging prediction discrepancy.

*Figure 1.* (a) shows that balanced generation leads to a mismatch with the inherently imbalanced global distribution. (b) shows that we observe when a class is underrepresented in a client's local dataset, the local model incurs a higher loss compared to the global model.

not only natural but also beneficial for model performance (Chou et al., 2021), which we provide a detailed discussion in Appendix B. For instance, in weather forecasting, some regions experience more rainy days than sunny days, and the global training datasets should reflect this imbalance to improve predictive accuracy. In such contexts, enforcing local class balance through synthetic data generation introduces a mismatch between training and testing distributions, which can degrade model performance. This challenge is further compounded in FL by strict privacy and communication constraints, which make it difficult for the server to accurately estimate the true global distribution. As a result, existing balance-oriented augmentation strategies become suboptimal in these settings. This raises a critical and underexplored question: **How can client-side data be synthesized and selected such that local distributions align with a true, yet unknown global distribution?**

To address the aforementioned challenge, we propose Fed-PDG, a novel framework that dynamically adjusts the data supplementation process according to the true global distribution, rather than assuming it to be balanced. The key insight behind FedPDG is that a model's behavior implicitly encodes the data distribution it has learned: when a model performs poorly on a given sample, it suggests that the sample lies outside the model's learned distribution.

Leveraging this insight, FedPDG quantifies the prediction gap between the global and local models on each generated sample, using this as a signal to guide selective supplementation. Specifically, the framework compares the losses incurred by the local and global models on each synthetic sample. If the local model yields a higher loss, it indicates that the client's local data distribution underrepresents this type of data. Such samples are then selectively allocated to

the client for training, thereby implicitly aligning local and global distributions without requiring access to or revealing the client's private data. Figure 1a illustrates a toy example contrasting traditional balance-oriented augmentation with our proposed method. Figure 1b explains how prediction discrepancy between local and global models reveals underrepresented classes in a client's dataset.

Beyond selective data supplementation, we aim to customize the diffusion model to generate data tailored to the needs of each client. To instill a desired generative preference in the model, we adopt a Direct Preference Optimization (DPO) paradigm (Rafailov et al., 2024) and train a lightweight LoRA module (Hu et al., 2021) for each client. We design a reward function, which encourages the diffusion model to generate samples with high client loss, low global loss, and disagreement between the client and global model predictions. This enables the generator to produce data that better captures what is lacking relative to the global distribution.

Importantly, FedPDG introduces no additional computational or communication overhead on the client side, making it both practical and efficient for real-world deployment. Furthermore, it is designed as a plug-and-play module that can be seamlessly integrated into standard FL pipelines, as well as heterogeneous FL settings involving diverse model architectures or resource-constrained clients.

We extensively evaluate FedPDG on CIFAR-10, CIFAR-100 and Tiny Imagenet datasets, showing that FedPDG consistently improves global model performance under non-IID conditions. Extensive experiments on multiple benchmarks demonstrate that FedPDG outperforms state-of-the-art methods, achieving up to 3.82% improvement.

## 2. Related Work

### 2.1. Data Heterogeneity in Federated Learning

Data heterogeneity, refers to the scenario where the data available to each client is not drawn from the same distribution, which is typical in real-world federated learning settings (G. et al., 2024). This results that local models diverge from the global model as they are trained on different data distributions. As a result, the global model may struggle to generalize well across all clients, reducing its performance and effectiveness in practice. To handle non-IID data effectively, some methods (Sahu et al., 2018; Karimireddy et al., 2019; Lee et al., 2024) introduce a regularization term or control variance to constrain local updates. Methods like (Li et al., 2021; Hsu et al., 2019; Fan et al., 2024) have employed momentum and other strategies to alleviate drift. These methods mainly rely on aligning local models with the global objective rather than directly addressing the underlying data heterogeneity.

Another line of work addresses data heterogeneity through personalization, maintaining client-specific models rather than a single global one. Many such methods decouple shared and personalized knowledge (Wu et al., 2024; 2023; Zhu et al., 2024; Wu et al., 2025b; 2026b; Qi et al., 2023; 2025a;b). Further studies tackle spatial-temporal heterogeneity (Wu et al., 2026a), OOD generalization (Zhang et al., 2025; 2026), and universal personalization with collaborative generative models (Wu et al., 2025a). Unlike these parameter or representation level approaches, our work directly reshapes the client data distributions.

### 2.2. Federated Generative Learning

Recently, several approaches (Abacha et al., 2024) in Federated Learning have been proposed to enhance model performance by using diffusion models to generate synthetic data. Gen-FedSD (Morafah et al., 2024) leverages Stable Diffusion to generate class-specific synthetic images on each client, with the goal of balancing label distribution and mitigating data heterogeneity in federated learning (FL). GenFL (Qiang et al., 2025) trains an auxiliary model solely on the generated data, and then aggregates it with the client models on the server side through weighted averaging. While these methods perform well when the global data distribution is balanced, they encounter significant challenges in scenarios where the global distribution is imbalanced, leading to mismatches between the client data distributions and the global distribution. DPSDA-FL (Abacha et al., 2024) uses foundation models to generate differentially private synthetic data locally, which is then shared and redistributed across clients to balance local data distributions. Astraea (Duan et al., 2019) exposes the client data distributions to the server, which computes the categories that each client should supplement. While this method can align the local distributions

with the global distribution in the case of an imbalanced global distribution, it reveals client label distributions to the server, which is impractical and raises significant privacy concerns.

Some approaches in one-shot Federated Learning also leverages synthetic data. Typically, these approaches (Zhang et al., 2024; Yang et al., 2024b;a; Chen et al., 2025) deploy a diffusion model on the client-side, where clients generate representative embeddings based on their local data. These embeddings are then uploaded to the server, which uses them to generate diverse synthetic images. However, directly transferring such methods into standard multi-round FL is non-trivial. To extract the representations in these methods, clients are typically required to run large-scale generative or vision-language models (e.g., diffusion models or BLIP-2) locally—an assumption that introduces substantial computational and memory overhead. While this may be acceptable in one-shot scenarios with a single round of communication, it becomes infeasible in regular FL settings, where the cost would be incurred repeatedly across rounds.

## 3. Methodology

### 3.1. Problem Definition

We consider a standard federated learning (FL) setting with $N$ clients $\mathcal{C} = \{1, 2, \ldots, N\}$. Each client $i \in \mathcal{C}$ owns a private dataset $\mathcal{D}_i = \{(x_j^{(i)}, y_j^{(i)})\}_{j=1}^{N_i}$ from its local data distribution $\mathcal{P}_i(x, y)$, which is not shared with other clients or the server. Let $\mathbf{d}_i = (d_{i,1}, d_{i,2}, \ldots, d_{i,C})$ denote the label distribution vector of client $i$, where $d_{i,c}$ denotes the proportion of samples from class $c$ in client $i$'s dataset. The global label distribution $\mathbf{d}_g = (d_{g,1}, d_{g,2}, \ldots, d_{g,C})$ is defined as:

$$d_{g,c} = \frac{\sum_{i=1}^{N} N_i \cdot d_{i,c}}{\sum_{i=1}^{N} N_i}, \quad c = 1, \ldots, C. \tag{1}$$

In our scenario, the local label distributions $\{\mathbf{d}_i\}_{i=1}^{N}$ are heterogeneous (non-IID across clients), and *the global label distribution $\mathbf{d}_g$ can also be inherently imbalanced*. The server aims to learn a global model $w_g$ that performs well under the unknown global label distribution $\mathbf{d}_g$. Formally, the objective can be written as maximizing the predictive accuracy:

$$w_g^* = \arg\max_{w} \ \mathbb{E}_{(x,y)\sim\mathcal{P}_g}\big[\mathbf{1}\{h_w(x) = y\}\big], \tag{2}$$

where $\mathcal{P}_g(x, y)$ denotes the global data distribution with label prior $\mathbf{d}_g$, and $h_w : \mathcal{X} \to \mathcal{Y}$ is the prediction function parameterized by $w$.

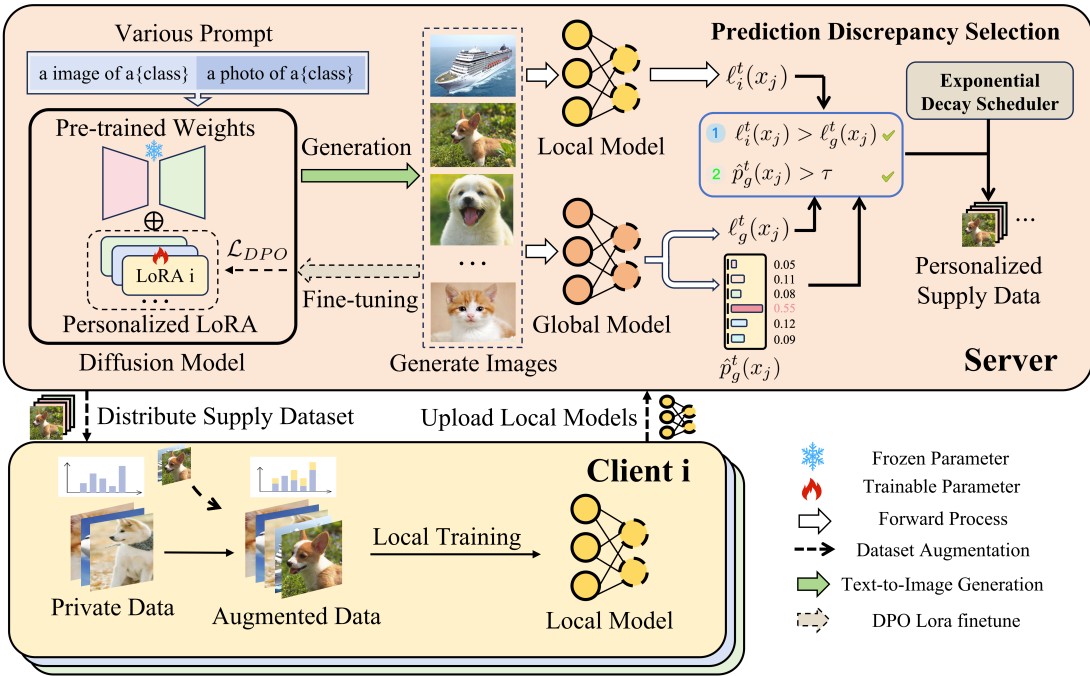

*Figure 2.* Overview of the FedPDG framework. The server generates synthetic images using a diffusion model and selects samples based on prediction discrepancy and global confidence, with the supply amount controlled by an Exponential Decay Scheduler. To further personalize generation, client-specific LoRA modules are fine-tuned via Discrepancy Preference Optimization (DPO). The selected synthetic samples are then distributed to clients, augmenting their private data and improving alignment with the global distribution.

## 3.2. Prediction Discrepancy Guided Data Selection

Though the global distribution $\mathbf{d}_g$ is unknown in practice, we show that the prediction discrepancy between a local model $w_i$ and the global model $w_g$ on a given sample $(x, y)$ can reveal whether client $i$'s label distribution $\mathbf{d}_i$ is relatively deficient in class $y$ compared to $\mathbf{d}_g$.

Let $\ell(w; x, c)$ denotes the loss of model $w$ on the sample $(x, c)$, $q(\cdot \mid c)$ denotes the class-conditional data distribution for label $c$. Following the Bayes' theorem (Bayes, 1763) and conclusion in (Gneiting & Raftery, 2007; Nguyen et al., 2010),we obtain this theorem (concrete proof in Appendix C):

**Theorem 3.1.** *If $d_{i,c} < d_{g,c}$, then client $i$ suffers a strictly larger expected loss than the global model on class $c$:*

$$\mathbb{E}_{x \sim q(\cdot|c)}\big[\ell(w_i; x, c) - \ell(w_g; x, c)\big] > 0. \tag{3}$$

**Basic Selection Mechanism**  Building on the conclusion above, we design a practical mechanism to leverage prediction discrepancy for data supplementation in federated learning. In each communication round $t$ of FL, client $i$ locally trains a model $w_i^t$ on its private dataset $\mathcal{D}_i$ and uploads the updated model to the server. The server then aggregates all client models to obtain the global model $w_g^t$. We employs a text-to-image diffusion model on the server. In each round $t$, server randomly samples from a predefined prompt template set to generate image set $\tilde{D}_i^t$. For each synthetic

sample $(x_j, y_j) \in \tilde{D}_i^t$, if:

$$\ell_i^t(x_j, y_j) - \ell_g^t(x_j, y_j) > 0 \tag{4}$$

it is assigned to client $i$ as supplemental data.

**Confidence-aware Filter**  To prevent the supplementation of low-quality or uninformative samples, we introduce a confidence-aware filtering criterion. Specifically, we require that the global model not only achieves a lower loss on a sample but also exhibits sufficient prediction confidence. We define $\hat{p}_g^t(x_j)$ as:

$$\hat{p}_g^t(x_j) = \text{softmax}(w_g^t(x_j))_{y_j}, \tag{5}$$

We require the confidence to exceed a random-guess probability $\tau = \frac{1}{C}$, where $C$ is the number of classes. The sample assignment rule is then updated as

$$(x_j, y_j) \in \tilde{\mathcal{D}}_i, \quad \text{if} \quad \ell_i^t(x_j, y_j) - \ell_g^t(x_j, y_j) > 0 \\ \text{and} \quad \hat{p}_g^t(x_j) > \tau \tag{6}$$

This additional constraint not only filters out low-quality samples but also stabilizes the optimization process: under cross-entropy loss, the gradient variance is bounded by $2(1 - \tau)$ (proof in Appendix D), thereby enhancing the stability of the supplementation process and preventing noisy gradients from destabilizing training.

**Exponential Decay scheduler**   Although the prediction discrepancy mechanism helps align local distributions with the global distribution, we claim the quantity of synthetic data supplied in each round remains a critical factor in the training process. In the early stages of training, the model possesses limited knowledge, and the discrepancy signal may not accurately reflect the true distributional gap. On the other hand, supplying too little data early on may cause the model to converge to a poor local minimum, making it difficult to correct even with larger supplementation in later rounds. Inspired by various learning rate schedulers (Konar et al., 2020), we experiment with four representative forms: uniform, linear decay, stepwise decay, and exponential decay. The exponential strategy follows a simple normalized power-law decay:

$$N_t = N_{total} \cdot \frac{1}{\sum_{s=1}^{T} s^{-\beta}} \cdot t^{-\beta}, \quad (7)$$

where $N_t$ is the number of samples supplied for each client, $N_{total}$ is the fixed total number of samples supplied for each client, $\beta > 0$ is a hyperparameter controlling the decay rate. The detailed scheduling and training details are provided in Appendix E. The empirical results are presented in Table 1, which demonstrate that the exponential decay scheduler achieves the best trade-off between early-stage acceleration and late-stage stability.

Our full pipeline is illustrated in Figure 2, which provides an overview of how synthetic samples are filtered and distributed to clients.

*Table 1.* Accuracy comparison of different scheduling strategies on CIFAR-10 Dataset.

| Scheduler | $\alpha = 0.1$ | $\alpha = 0.5$ | $\alpha = 1.0$ |
|---|---|---|---|
| Uniform | 52.07 | 51.35 | 50.82 |
| Linear | 53.75 | 55.14 | 56.28 |
| Stepwise | 55.56 | 55.98 | 56.45 |
| **Exponential** | **56.16** | **56.10** | **57.48** |

### 3.3. Discrepancy Preference Optimization

The selection mechanism described above effectively aligns each client's label distribution $\mathbf{d}_i$ with the global label distribution $\mathbf{d}_g$. Building on this foundation, we further adapt the diffusion model itself to enable personalized data generation, ensuring that each client receives synthetic samples tailored to its specific needs. Instead of acquiring entirely new knowledge, the diffusion model primarily needs to *adjust its generation bias toward the regions of the data space where each client is under-represented.*

To achieve this, we introduce the Discrepancy Preference Optimization paradigm, which leverages implicit preference

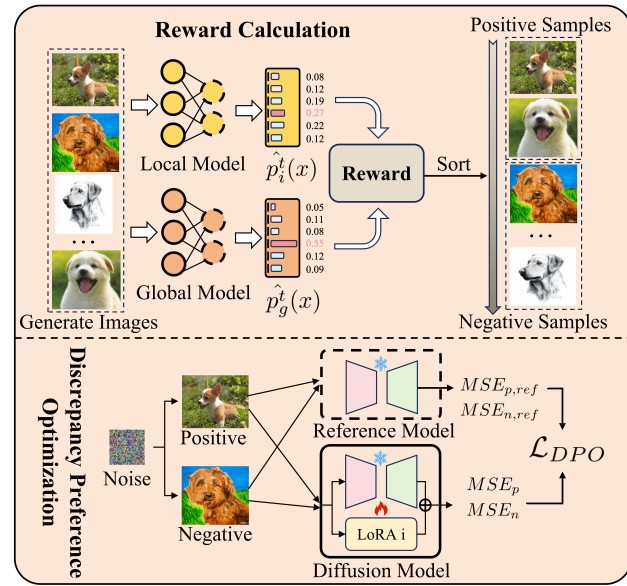

*Figure 3.* Illustration of Discrepancy preference optimization module.

signals to guide lightweight LoRA finetuning. Figure 3 provides an illustration of this module. Recall in each round $t$, the server uses diffusion model to generate synthetic images. For each sample $(x_j, y_j)$, we compute reward for each client $i$ as:

$$r_{i,j}^t = \hat{p}_g^t(x_j) - \hat{p}_i^t(x_j) \ + \ \lambda \cdot disagree(x_j), \quad (8)$$

where $\hat{p}_i^t(x_j)$ and $\hat{p}_g^t(x_j)$ are defined in Equation 5, and *disagree* term is a binary indicator that equals 1 if and only if the global model and the local model predict different classes for $x_j$, and 0 otherwise.

$$disagree(x_j) = \mathbf{1}\{h_{w_i^t}(x_j) \neq h_{w_g^t}(x_j)\}. \quad (9)$$

The motivation behind this reward function is twofold. First, we encourage samples that are difficult for the client but easy for the global model, as this implies that such patterns are under-represented in the client's data distribution. Second, we encourage samples on which the client and global models make different predictions, since disagreement indicates regions of distributional mismatch where supplemental data can most effectively reduce divergence.

Based on the reward in Equation 8, we sort the generated samples for client $i$ in descending order. The top half of the $\tilde{D}_i^t$ samples are regarded as *preferred* samples, while the bottom half are regarded as *rejected* samples. We then form preference pairs $\mathcal{S}_i^t = \{(x^+, x^-, y)\}$, where $x^+$ comes from the preferred set and $x^-$ from the rejected set with the same label $y$.

For each pair, let $\pi_\theta(x \mid y)$ denotes the likelihood under the current diffusion model with client-specific LoRA parameters $\theta$, and $\pi_{ref}(x \mid y)$ denotes the likelihood under the

*Table 2.* Comparison of federated methods on CIFAR-10 and CIFAR-100 dataset.

| Method | CIFAR-10 | | | | CIFAR-100 | | |
|---|---|---|---|---|---|---|---|
| | $\alpha = 0.05$ | $\alpha = 0.1$ | $\alpha = 0.5$ | $\alpha = 1.0$ | $\alpha = 0.1$ | $\alpha = 0.5$ | $\alpha = 1.0$ |
| FedAvg (2017) | 51.58±2.46 | 51.36±3.82 | 50.60±2.89 | 52.98±0.99 | 34.19±1.18 | 29.63±2.02 | 27.73±2.21 |
| FedProx (2018) | 52.99±2.41 | 50.60±4.44 | 50.50±2.79 | 52.32±1.77 | 35.42±1.75 | 28.81±1.97 | 27.23±1.83 |
| FedProto (2022) | 52.18±2.25 | 50.38±4.09 | 50.10±2.92 | 54.02±1.54 | 34.87±1.32 | 29.39±1.68 | 27.61±1.30 |
| FedETF (2023) | 47.86±1.13 | 48.70±3.97 | 48.10±3.79 | 49.69±1.05 | 35.45±1.92 | 28.97±1.23 | 26.60±0.82 |
| FedFA (2023) | 45.43±2.16 | 47.67±3.42 | 50.24±1.98 | 52.16±1.58 | 32.37±3.08 | 28.46±1.61 | 26.21±1.98 |
| DPSDA-FL (2024) | 53.17±0.80 | 53.29±4.26 | 52.40±3.18 | 55.94±0.81 | 35.80±1.30 | 30.40±1.50 | 28.90±1.60 |
| CRFed (2024) | 61.28±1.95 | 56.84±3.21 | 54.72±2.48 | 55.83±1.31 | 36.52±1.47 | 31.86±1.62 | 30.24±1.38 |
| GenFL (2025) | 54.45±1.45 | 52.12±4.00 | 51.44±2.91 | 53.94±0.69 | 35.24±1.23 | 29.08±1.88 | 27.40±1.84 |
| Gen-FedSD (2024) | 63.36±1.80 | 58.30±3.73 | 53.27±2.70 | 56.20±1.52 | 36.80±1.60 | 31.09±1.04 | 29.68±1.77 |
| **FedPDG** | **64.64±2.23** | **59.96±3.67** | **57.09±3.14** | **58.95±1.37** | **37.41±1.12** | **32.44±1.75** | **31.58±1.45** |

reference diffusion model (a copy of pretrained diffusion model, without LoRA). Define the pairwise logit differences as

$$\Delta_\theta = \log \pi_\theta(x^+ \mid y) - \log \pi_\theta(x^- \mid y) \tag{10}$$

$$\Delta_{\text{ref}} = \log \pi_{\text{ref}}(x^+ \mid y) - \log \pi_{\text{ref}}(x^- \mid y). \tag{11}$$

Following the Direct Preference Optimization (DPO) paradigm, the training objective for client $i$ at round $t$ is

$$\mathcal{L}_{DPO}(\theta; i, t) = -\frac{1}{|\mathcal{S}_i^t|} \sum_{(x^+, x^-, y) \in \mathcal{S}_i^t} \log \sigma \Big( \kappa(\Delta_\theta - \Delta_{\text{ref}}) \Big), \tag{12}$$

where $\sigma(\cdot)$ is the logistic function and $\kappa > 0$ controls the sharpness of preference alignment. The training details are shown in Appendix F.

## 4. Experiment and Analysis

### 4.1. Experimental Settings

**Networks and Datasets.** Following prior studies, we evaluate the performance of our method and several state-of-the-art baselines on CIFAR-10 (Krizhevsky et al., 2010), CIFAR-100 (Krizhevsky, 2009), and Tiny-ImageNet (Le & Yang, 2015). We utilize Stable Diffusion v1-4 to generate synthetic data for clients and adopt ResNet (He et al., 2016) as each client's local model. For CIFAR-10, we use the ResNet-8 model, each client having 500 samples. For CIFAR-100, we use the ResNet-10 model, with 1000 samples per client. For Tiny-ImageNet, we use the ResNet-10 model, with 1500 samples per client. To simulate globally imbalanced scenarios under different levels of heterogeneity, we modify the test data distribution to approximate the global label distribution, with detailed construction in Appendix I

**Baselines and Implementation.** We compare our proposed method against several representative baselines. All

*Table 3.* Comparison of federated methods on the Tiny-ImageNet dataset across different heterogeneity levels $\alpha$.

| Method | $\alpha = 0.1$ | $\alpha = 0.5$ | $\alpha = 1.0$ |
|---|---|---|---|
| FedAvg | 20.94±0.80 | 15.41±1.06 | 15.61±0.33 |
| FedProx | 21.60±0.70 | 15.63±1.24 | 16.30±0.29 |
| FedProto | 22.07±0.60 | 15.68±0.83 | 15.88±0.48 |
| FedETF | 19.99±0.99 | 14.90±1.19 | 13.94±0.27 |
| FedFA | 21.21±0.44 | 13.49±1.10 | 13.45±0.47 |
| DPSDA-FL | 13.30±0.25 | 15.10±0.30 | 15.45±0.40 |
| CRFed | 22.43±0.45 | 16.72±0.38 | 16.98±0.41 |
| GenFL | 12.39±0.36 | 14.26±0.28 | 14.45±0.43 |
| Gen-FedSD | 13.88±0.11 | 15.57±0.17 | 15.96±0.41 |
| **FedPDG** | **23.27±0.37** | **17.56±0.42** | **17.75±0.43** |

methods are trained for 500 communication rounds using the SGD optimizer with a fixed learning rate of 0.1. For Generative methods which generates balanced synthetic data for clients, we use the same predefined prompt set as ours to generate images. We also ensure that the total amount of synthetic data supplied per client for these baselines matches that of our method to make a fair comparison.

Specifically, in all scenarios, the quantity of generated data per client is set equal to the size of its initial local dataset. For our method, FedPDG, we set the supply coefficient $\beta$ to 1.0 for CIFAR-10 and Tiny-ImageNet datasets, and 1.25 for CIFAR-100. In the experiments on CIFAR-10, we select the Dirichlet coefficient $\alpha$ from $\{0.05, 0.1, 0.5, 1.0\}$. and $\{0.1, 0.5, 1.0\}$ for CIFAR-100 and Tiny-Imagenet. The text-to-image generation details are shown in Appendix G.

For each dataset and heterogeneity setting, we run each method three times with different random seeds and report the mean accuracy along with the standard deviation in Table 2 and Table 3, with statistical significance test results in Appendix H.3. We record the best accuracy achieved by

the global model during training.

## 4.2. Case Study

In this section, we present a case study conducted using the CIFAR-10 dataset with $\alpha = 1.0$ under an imbalanced global distribution scenario. This case study is designed to demonstrate the effectiveness of our method in aligning local and global distributions from multiple perspectives.

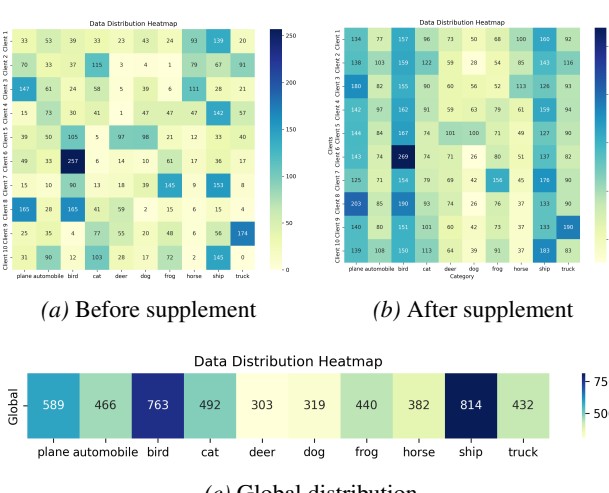

*(a)* Before supplement     *(b)* After supplement

*(c)* Global distribution

*Figure 4.* Data Distribution Heatmap

We can observe that the distribution in Figure 4b closely resembles that of Figure 4c. Then, we take Client 2 as an example. Under a balanced generative supplementation strategy, the algorithm would tend to prioritize the most underrepresented classes—such as classes 5, 6, and 7, which would result in a mismatch with the global distribution. In contrast, our framework instead supplements a large amount of data for class 2, which is also one of the most abundant classes in the global distribution. This demonstrates that our method effectively avoids misleading local supplementation and better aligns with the global data trend.

Beyond the qualitative case study, we further conduct a quantitative analysis on CIFAR-10 under different heterogeneity levels. Specifically, we evaluate the average KL divergence between each client's post-supplementation distribution and the true global distribution. We compare four strategies: the original distribution without supplementation, random supplementation, rare-class supplementation (prioritizing the most underrepresented classes, as in Gen-FedSD), and our proposed method. As shown in Table 4, our method consistently achieves the lowest KL divergence across all settings, substantially reducing the discrepancy between local and global distributions.

*Table 4.* Average KL divergence for different data-supply strategies (lower is better).

| Method | $\alpha = 0.05$ | $\alpha = 0.1$ | $\alpha = 0.5$ | $\alpha = 1.0$ |
|---|---|---|---|---|
| Origin | 1.376 | 1.416 | 0.656 | 0.359 |
| Random | 0.453 | 0.658 | 0.185 | 0.107 |
| Rare-class | 0.350 | 0.651 | 0.097 | 0.049 |
| **Ours** | **0.251** | **0.291** | **0.094** | **0.038** |

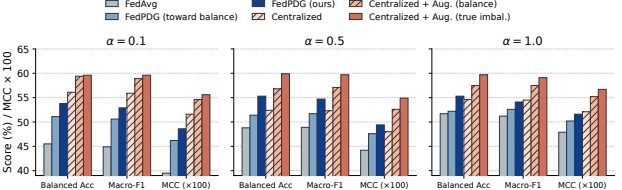

*Figure 5.* Balanced Accuracy, Macro-F1 and MCC under imbalanced distribution on CIFAR-10.

## 4.3. Analysis

**Evaluation under Imbalanced-Classification Metrics**
To prevent inflated accuracy caused by model collapse to the majority class, we conduct experiments on CIFAR-10 with three additional metrics: *Balanced Accuracy*, *Macro-F1*, and the *Matthews Correlation Coefficient (MCC)*. We also include two centralized references trained on the union of all clients' data: a vanilla model, and one with class-balancing augmentation (*toward balance*) or augmentation matching the true imbalance (*toward true imbalance*). Results are reported in Figure 5.

The results show that our method improves performance across all classes, not merely the frequent ones.

**Client-side Computational overhead Analysis.** Although our method supplements clients with additional synthetic data, we adopt only one local training epoch per communication round, in contrast to the five epochs used by other baselines. To quantify the effect, we calculate the total training FLOPs of ResNet-8 under different dataset sizes. As shown in Table 5, our method consistently requires less computation compared to FedAvg, achieving about 39.8% of the baseline cost.

*Table 5.* Quantitative computational analysis for clients (TFLOPS).

| | 500 samples | 1000 samples | 1500 samples |
|---|---|---|---|
| FedAvg | 111.6 | 222.3 | 334.8 |
| **Ours** | **44.5** | **88.4** | **132.9** |
| Ratio | 0.3993 | 0.3977 | 0.3972 |

**Server-side Computational overhead Analysis** We analyze server-side computational overhead on A100 GPUs. As shown in Figure 6(a), the overhead consists of three stages:

image generation (0.40%), model inference (<0.01%), and DPO training (4.65%), totaling approximately 5% of end-to-end runtime. Figure 6(b) shows that due to the exponential decay scheduler, image generation time decreases from 37.19s to 8.13s within five epochs. After epoch 4, generation time falls below the average client training time, meaning the server completes data preparation before clients finish local training, thus introducing no additional waiting time in the FL pipeline. This demonstrates that FedPDG is computationally feasible for practical deployment.

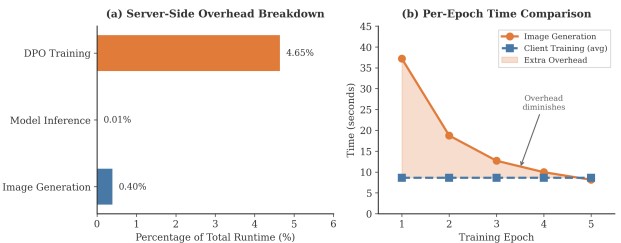

*Figure 6.* Server-side computational overhead analysis on A100 GPUs.

**Computationally Efficient Variant of FedPDG** We further scale our method and observe that the DPO computation overhead grows linearly with the number of clients, i.e., $O(N)$, shown in Figure 7(a). To adapt our method to scenarios with extremely large $N$, we introduce *Computationally Efficient FedPDG*, which trains a single shared LoRA module across all clients using the (generated sample, reward) pairs collected from each client. The samples are sorted by reward values, and we retain the same total number of samples as the original per-client DPO training to match the computation budget. As shown in Figure 7, this variant has a little performance drop but still outperforms the best baseline. Details are shown in Appendix H.

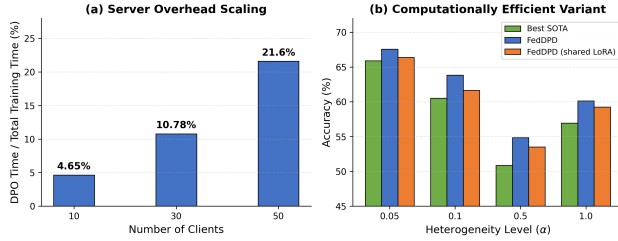

*Figure 7.* (a): Server overhead with scaling. (b): Results of Share LoRA variant.

**Domain shift of the diffusion model** To evaluate whether FedPDG generalizes to domains underrepresented in the pretraining data of Stable Diffusion, we conduct additional experiments on the satellite imagery domain (EuroSAT). Satellite domain differ substantially from the natural images used during the generator's pretraining.

As shown in Table 6, FedPDG consistently outperforms

all baselines under high class-imbalance scenarios ($\alpha \leq 0.5$). In mild imbalance settings ($\alpha = 1.0$), performance slightly degrades as the domain gap between synthetic and real data becomes the dominant factor while the benefit of addressing heterogeneity diminishes. This limitation can be mitigated by replacing the general-purpose diffusion model with domain-specific generators.

*Table 6.* Performance comparison on EuroSAT.

| Method | 0.05 | 0.1 | 0.5 | 1.0 |
|---|---|---|---|---|
| FedAvg | 51.74 | 61.92 | 82.09 | 85.94 |
| FedFA | 59.04 | 75.89 | 89.23 | 90.24 |
| FedProto | 58.99 | 77.19 | 90.98 | **90.50** |
| GenFL | 59.13 | 74.51 | 86.57 | 81.20 |
| Gen_SD | 63.29 | 78.40 | 87.14 | 85.33 |
| **FedPDG** | **67.30** | **82.46** | **91.10** | 87.62 |

**Plug-and-Play Framework.** In this section, we explore the potential of combining our proposed FedPDG framework with traditional FL methods. We conduct experiments combining FedPDG with FedProx, FedFA on the CIFAR-100 dataset with 10 clients, each having 1000 samples. We combine FedPDG with FedETF on the CIFAR-10 dataset, 10 clients having 500 samples. The experimental results with different dirichlet coefficient $\alpha$ is displayed in Table 7 and Table 8, which demonstrate an improvement over the single method.

*Table 7.* Performance of combining our method (FedPDG) with FedProx and FedFA.

| Method | $\alpha = 0.1$ | $\alpha = 0.5$ | $\alpha = 1.0$ |
|---|---|---|---|
| FedProx | 34.17 | 29.89 | 27.93 |
| **+FedPDG** | **36.38** | **34.65** | **32.00** |
| FedFA | 37.61 | 35.82 | 33.16 |
| **+FedPDG** | **39.39** | **36.07** | **35.20** |

*Table 8.* Performance of combining our method (FedPDG) with FedETF.

| Method | $\alpha = 0.05$ | $\alpha = 0.1$ | $\alpha = 0.5$ | $\alpha = 1.0$ |
|---|---|---|---|---|
| FedETF | 51.11 | 44.49 | 46.48 | 53.48 |
| **+FedPDG** | **58.67** | **52.37** | **56.94** | **59.49** |

**Domain applicability** Notably, FedPDG can adapt well on different domains by choosing a suitable generative model. Currently, domain-specific diffusion models are increasingly available and can be substituted directly, including generators for chest CT (Hamamci et al., 2024), X-ray (Xie et al., 2025), satellite imagery (Khanna et al., 2024), and fine-grained species (Monsefi et al., 2025); consequently, the regime lacking an adequate generator is narrow,

essentially limited to domains requiring highly specialized priors (e.g. molecular structures or circuit schematics). We also conduct experiments to validate this, shown in Appendix J.

### 4.4. Ablation Study

We perform an ablation study on the CIFAR-10 dataset to evaluate the impact of three key components of our method: the Prediction Discrepancy Selection (PDS), the Exponential Supply Scheduler (ESS) and Discrepancy Direct Preference Optimization (DPO).

*Table 9.* Ablation study of different components in our FedPDG framework on CIFAR-10.

| PDS | ESS | DPO | $\alpha = 0.1$ | $\alpha = 0.5$ | $\alpha = 1.0$ |
|-----|-----|-----|----------------|----------------|----------------|
|     |     |     | 46.24 | 48.46 | 51.92 |
| ✓   |     |     | 52.07 | 51.35 | 55.82 |
| ✓   | ✓   |     | 54.35 | 54.53 | 56.79 |
|     |     | ✓   | 49.00 | 52.49 | 55.04 |
| ✓   | ✓   | ✓   | **55.44** | **54.90** | **57.03** |

The experimental results are presented in Table 9. For Fed-PDG (w/ PDS), we adopt a default setting where each client receives the same number of random samples per communication round. For FedPDG (w/ PDS + ESS), the number of selected samples is determined by the ESS scheduling rule. For FedPDG (w/ DPO), neither selection nor scheduling is applied. The results show that all three components contribute to improving the global model.

### 4.5. Hyperparameter Sensitivity

To evaluate the sensitivity of FedPDG to hyperparameters, we conduct experiments varying the key parameters on CIFAR-10 with $\alpha = 0.1$. Specifically, we examine the exponential decay rate $\beta$, the confidence threshold $\tau$, and the DPO reward weight $\lambda$. As shown in Figure 8, performance remains stable across a wide range of hyperparameter values, with accuracy fluctuating within approximately 2% around the optimal setting ($\beta = 1.0$, $\tau = 0.1$, $\lambda = 0.5$). Since $\tau$ varies across datasets, we provide additional experiments on different datasets in Table 16. This demonstrates that FedPDG is robust to hyperparameter choices and does not require extensive tuning for practical deployment.

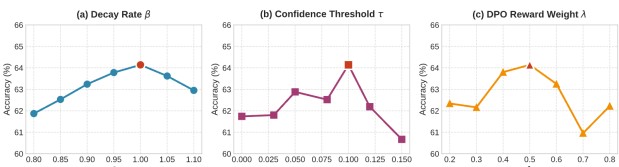

*Figure 8.* Hyperparameter sensitivity analysis.

### 5. Conclusion

We introduce FedPDG, a synthetic-data framework that leverages prediction discrepancy to align local and global distributions—without assuming global balance—and thereby improves robustness under class imbalance and non-IID settings on CIFAR-10, CIFAR-100 and Tiny-ImageNet. Future work will further evaluate FedPDG in larger-scale real-world federated deployments and explore stronger domain-specific generators for highly specialized domains.

### Impact Statement

This paper presents work whose goal is to advance the field of Machine Learning. There are many potential societal consequences of our work, none which we feel must be specifically highlighted here.

### Acknowledgement

This work was supported by the National Natural Science Foundation of China under Grants 62372028 and 62372027, and by the Central Guiding Local Science and Technology Development Fund of Shanghai Municipality (Project No. YDZX20253100004011)

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

# A. Pseudo Code

---

**Algorithm 1** FedPDG Framework

---

1: **Input:** pretrained diffusion model $\theta$, local dataset $\mathcal{D}_i$, number of rounds $T$, exponential decay rate $\beta$, confidence threshold $\tau = \frac{1}{C}$, maximum supply data amount $N_{\text{total}}$ for each client.
2: **Output:** Updated global model $w_g^T$
3: Initialize global model $w_g^0$, local models $w_i^0$ for each client $i$
4: Set $t = 0$
5: **while** $t \leq T$ **do**
6:     {Client executes:}
7:     Train local model: $w_i^t \leftarrow \texttt{LocalTrain}(w_i^t, \mathcal{D}_i)$
8:     Upload local models $w_i^t$ to the server
9:     {Server executes:}
10:     Server aggregates local models to get global model $w_g^t$
11:     Calculate Supply Amount: $N_t = N_{\text{total}} \cdot \frac{1}{\sum_{s=1}^{T} s^{-\beta}} \cdot t^{-\beta}$
12:     **for** each client $i$ **do**
13:         Set $count = 0$
14:         **while** $count < N_t$ **do**
15:             Generate synthetic sample $(x_j, y_j) \sim \theta$
16:             Compute losses $\ell_g^t, \ell_i^t$, discrepancy $s_{i,j}$, and confidence $\hat{p}_g^t$
17:             **if** $s_{i,j} > 0$ **and** $\hat{p}_g^t(x_j) > \tau$ **then**
18:                 Assign $(x_j, y_j)$ to client $i$: $\tilde{\mathcal{D}}_i \leftarrow \tilde{\mathcal{D}}_i \cup \{(x_j, y_j)\}$
19:                 $count = count + 1$
20:             **end if**
21:         **end while**
22:         Distribute $\tilde{\mathcal{D}}_i$ to client $i$
23:     **end for**
24:     Server trains LoRA for clients: $\texttt{DPO Training}(w_i^t, w_g^t, \tilde{\mathcal{D}})$
25:     {Client executes:}
26:     Update local model $w_i^t = w_g^t$
27:     Merge Dataset: $\mathcal{D}_i = \mathcal{D}_i \cup \tilde{\mathcal{D}}_i$
28:     $t = t + 1$
29: **end while**
30: **return** $w_g^T$

---

Algorithm 1 summarizes the complete FedPDG framework. At each communication round $t$, clients first train their local models on private data and upload them to the server (Lines 7–8). The server then aggregates the local models to obtain the global model and calculates the number of synthetic samples to supply based on the exponential decay scheduler (Lines 10–11).

For each client, the server generates synthetic samples using the pretrained diffusion model and evaluates them using the prediction discrepancy criterion: a sample is assigned to client $i$ only if the local model incurs a higher loss than the global model ($s_{i,j} > 0$) and the global model exhibits sufficient confidence ($\hat{p}_g^t(x_j) > \tau$) (Lines 12–22). After distributing the selected samples, the server performs Discrepancy Preference Optimization to fine-tune client-specific LoRA modules (Line 24). Finally, clients update their local models with the aggregated global model and merge the received synthetic data into their local datasets for subsequent training (Lines 26–28).

# B. When to Preserve vs. Mitigate Global Class Imbalance

A reasonable question is whether matching a globally imbalanced distribution is itself desirable. Actually it is *application-dependent*, and FedPDG targets the regime where preservation is appropriate.

**When mitigation is preferred (asymmetric-cost regime).** In safety-critical domains such as medical diagnosis, the cost of a false negative (missed cancer) far exceeds that of a false positive. The deployed model must be *deliberately* biased toward the minority class, even at the expense of overall accuracy. Class rebalancing (resampling, reweighting, augmentation toward balance) is the right tool here.

**When preservation is preferred (equal-cost regime).** Many federated deployments operate under approximately symmetric misclassification costs, where the goal is to maximize correctness on the natural test distribution:

- **E-commerce product classification.** Across federated merchants, categories such as phone cases vastly outnumber niche items. Misclassifying a phone case as a charger carries no special cost premium over misclassifying a vinyl record as a CD. Maximizing overall correctness is best served by training on the true product distribution.

- **Wildlife and plant species recognition.** Federated systems aggregating images from camera traps or nature apps naturally observe far more common species (deer, sparrows) than rare ones. For ecological surveys evaluated on *submitted* photos, the model should reflect the true species frequency.

- **Web image content classification.** Across heterogeneous web platforms, category frequencies are inherently skewed and reflect the true composition of web data. When costs are roughly symmetric, preserving this distribution is more appropriate than artificially equalizing it.

FedPDG is designed for this second regime. In the asymmetric-cost regime, FedPDG can be combined with cost-sensitive reweighting on top of its alignment mechanism; we leave a thorough treatment to future work.

## C. Proof of Prediction Discrepancy as an Indicator of Label Scarcity

**Assumptions.** All clients share the class-conditional densities $q(x \mid y)$, while only the class priors differ: $\mathcal{P}_i(x, y) = d_{i,y}\, q(x \mid y)$ and $\mathcal{P}_g(x, y) = d_{g,y}\, q(x \mid y)$. Let $m_i(x) = \sum_{k=1}^{C} d_{i,k}\, q(x \mid k)$ and $m_g(x) = \sum_{k=1}^{C} d_{g,k}\, q(x \mid k)$. We use cross-entropy loss $\ell(w; x, y) = -\log p_w(y \mid x)$.

**Step 1: Bayes posteriors for every $x$.** By Bayes' theorem, for every $x$ and $y$,

$$p_i(y \mid x) = \frac{d_{i,y}\, q(x \mid y)}{\sum_{c=1}^{C} d_{i,c}\, q(x \mid c)} = \frac{d_{i,y}\, q(x \mid y)}{m_i(x)}, \qquad p_g(y \mid x) = \frac{d_{g,y}\, q(x \mid y)}{\sum_{c=1}^{C} d_{g,c}\, q(x \mid c)} = \frac{d_{g,y}\, q(x \mid y)}{m_g(x)}. \tag{13}$$

**Step 2: Strictly proper scoring rules imply Bayes-consistency.** Cross-entropy (log loss) is a strictly proper scoring rule (Gneiting & Raftery, 2007; Nguyen et al., 2010). Hence, under sufficient model capacity and optimization convergence, minimizing the cross-entropy risk on $\mathcal{P}_i$ yields the Bayes-optimal predictor:

$$p_{w_i}(\cdot \mid x) = \arg\min_{r(\cdot \mid x) \in \Delta_C} \mathbb{E}_{(x,y) \sim \mathcal{P}_i}\big[-\log r(y \mid x)\big] = p_i(\cdot \mid x) \quad \text{a.s.} \tag{14}$$

An analogous statement holds for the global model, $p_{w_g}(\cdot \mid x) = p_g(\cdot \mid x)$.

**Step 3: Expected discrepancy (and strict positivity under overlap).** Fix a class $c$ and write $q_c(\cdot) \equiv q(\cdot \mid c)$. Using the Bayes posteriors from Step 1,

$$\ell(w_i; x, c) - \ell(w_g; x, c) = -\log p_{w_i}(c \mid x) + \log p_{w_g}(c \mid x) = \log \frac{p_g(c \mid x)}{p_i(c \mid x)}$$
$$= \log \frac{d_{g,c}}{d_{i,c}} + \log \frac{m_i(x)}{m_g(x)}. \tag{15}$$

Taking expectation over $x \sim q_c$ and using $\mathrm{KL}(q_c \| m) = \mathbb{E}_{q_c}[\log q_c(x) - \log m(x)]$, we obtain

$$\mathbb{E}_{x \sim q(\cdot \mid c)}\big[\ell(w_i; x, c) - \ell(w_g; x, c)\big] = \log \frac{d_{g,c}}{d_{i,c}} - \mathrm{KL}(q_c \| m_g) + \mathrm{KL}(q_c \| m_i). \tag{16}$$

Now write $m_\alpha(x) = \alpha\, q_c(x) + (1 - \alpha)\, r(x)$ with some density $r$ supported on $\{q(\cdot \mid k)\}_{k \neq c}$. Define $F(\alpha) = \mathrm{KL}\big(q_c \| m_\alpha\big)$. A direct calculation gives

$$F'(\alpha) \;=\; -\int q_c(x)\, \frac{q_c(x) - r(x)}{\alpha q_c(x) + (1 - \alpha) r(x)}\, dx \;\leq\; 0, \tag{17}$$

with strict inequality whenever $q_c$ and $r$ overlap on a set of positive measure. Hence $F(\alpha)$ is nonincreasing in $\alpha$. If $d_{g,c} > d_{i,c}$ then $\mathrm{KL}(q_c \| m_g) \leq \mathrm{KL}(q_c \| m_i)$, and

$$\mathbb{E}_{x \sim q(\cdot \mid c)}\big[\ell(w_i; x, c) - \ell(w_g; x, c)\big] \;\geq\; 0. \tag{18}$$

Moreover, if the class-conditionals are not perfectly separable (i.e., $q(\cdot \mid c)$ overlaps with some $q(\cdot \mid k)$, $k \neq c$), the inequality is strict:

$$\mathbb{E}_{x \sim q(\cdot \mid c)}\big[\ell(w_i; x, c) - \ell(w_g; x, c)\big] \;>\; 0. \tag{19}$$

**Conclusion.** Under label scarcity in client $i$ ($d_{i,c} < d_{g,c}$), the expected cross-entropy loss of the client model on class $c$ exceeds that of the global model when class-conditionals exhibit nontrivial overlap; in the degenerate perfectly separable case, the expectation equals $0$.

## D. Proof of gradient upper bound for confidence-aware filter

**Lemma D.1** (Per-sample loss bound). *Under cross-entropy loss with softmax probabilities $p_g^t(\cdot \mid x)$ and the confidence-aware filter $\hat{p}_g^t(x)_y \geq \tau$, we have*

$$\ell_g^t(x, y) = -\log \hat{p}_g^t(x)_y \;\leq\; -\log \tau.$$

*Proof.* Immediate from $\hat{p}_g^t(x)_y \geq \tau$ and the definition of cross-entropy. $\qquad\square$

**Lemma D.2** (Logit-gradient norm bound). *Let $z = w_g^t(x) \in \mathbb{R}^{\mathcal{C}}$ be the logits and $p = \mathrm{softmax}(z)$. For softmax cross-entropy $\ell(z, y) = -\log p_y$,*

$$\nabla_z \ell = p - e_y, \quad \text{hence} \quad \|\nabla_z \ell\|_2^2 = (1 - p_y)^2 + \sum_{k \neq y} p_k^2 \;\leq\; 2(1 - p_y). \tag{20}$$

*Under the filter $p_y \geq \tau$, it follows that*

$$\|\nabla_z \ell\|_2^2 \;\leq\; 2(1 - \tau), \qquad \|\nabla_z \ell\|_2 \;\leq\; \sqrt{2(1 - \tau)}. \tag{21}$$

*Proof.* Since $\sum_{k \neq y} p_k = 1 - p_y$ and $0 \leq p_k \leq 1$, we have $\sum_{k \neq y} p_k^2 \leq \sum_{k \neq y} p_k = 1 - p_y$. Therefore $\|\nabla_z \ell\|_2^2 \leq (1 - p_y)^2 + (1 - p_y) \leq 2(1 - p_y)$ because $(1 - p_y)^2 \leq (1 - p_y)$ for $p_y \in [0, 1]$. Applying $p_y \geq \tau$ yields the stated bounds. $\qquad\square$

**Lemma D.3** (Parameter-gradient bound). *Let $\theta$ denote the parameters of $w_g^t$ and suppose the logit Jacobian is bounded as $\|J_\theta z(x)\|_{\mathrm{op}} \leq L$ for all $x$ in the filtered set. Then, for filtered samples ($p_y \geq \tau$),*

$$\|\nabla_\theta \ell\|_2 = \|J_\theta z(x)^\top \nabla_z \ell\|_2 \;\leq\; L\, \|\nabla_z \ell\|_2 \;\leq\; L\sqrt{2(1 - \tau)}. \tag{22}$$

*Consequently,*

$$\mathbb{E}\big[\|\nabla_\theta \ell\|_2^2 \,\big|\, \hat{p}_g^t(x)_y \geq \tau\big] \;\leq\; 2L^2(1 - \tau). \tag{23}$$

*Proof.* Chain rule and the previous lemma. $\qquad\square$

**Corollary D.4** (Variance control). *Let $g = \nabla_\theta \ell$ for a filtered sample. Then $\mathrm{Var}(g) \leq \mathbb{E}\|g\|_2^2 \leq 2L^2(1 - \tau)$. In particular, increasing $\tau$ monotonically tightens the variance bound.*

# E. Details of different scheduler strategies

In this subsection, we provide the details of different scheduler strategies used for controlling the amount of synthetic data supplied to clients across communication rounds. Let $T$ denote the total number of communication rounds, $N_t$ denote the number of supplemental samples assigned to each client at round $t$ and $N_{\text{total}}$ is the total number of supplemental samples assigned to each client during training.

**Uniform scheduler.** Each client receives the same number of supplemental samples in every round:

$$N_t = \frac{N_{\text{total}}}{T}, \quad \forall t = 1, \dots, T, \tag{24}$$

**Linear decay scheduler.** The number of supplemental samples decreases linearly with the round index:

$$N_t = N_{\text{total}} \cdot \frac{T - t + 1}{\sum_{s=1}^{T} s}. \tag{25}$$

**Exponential scheduler.** The number of supplemental samples follows a normalized power-law decay:

$$N_t = N_{\text{total}} \cdot \frac{1}{\sum_{s=1}^{T} s^{-\beta}} \cdot t^{-\beta}, \tag{26}$$

where $\beta > 0$ controls the decay rate, and the normalization term ensures that $\sum_{t=1}^{T} N_t = D_{\text{total}}$.

We conduct comparison experiments on the CIFAR-10 dataset. The dataset is partitioned across 10 clients, with each client holding 500 local training samples. To evaluate the effectiveness of different supply scheduling strategies, we set the total number of supplemental samples to 500, which are distributed to clients within the first 40 communication rounds. The visualization of the per-round supplemental allocation is shown in Figure 9, which clearly illustrates the distribution pattern of each scheduler.

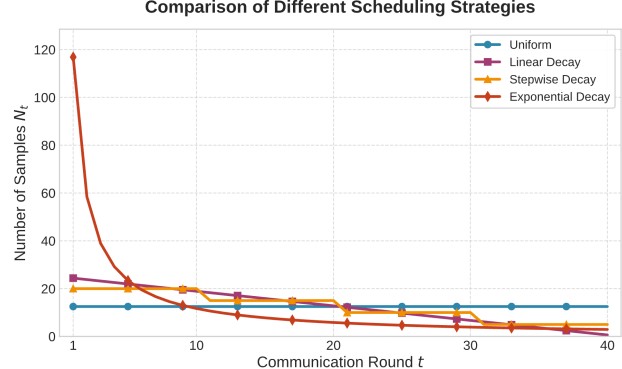

*Figure 9.* Comparison of sample supply $N_t$ across different scheduling strategies.

*Table 10.* Accuracy comparison of different generative backbones on CIFAR-10.

| Generative Model | $\alpha = 0.1$ | $\alpha = 0.5$ | $\alpha = 1.0$ |
|---|---|---|---|
| Stable Diffusion v1-4 | 36.32 | **34.46** | 31.93 |
| Flux-Dev | **37.98** | 34.40 | **32.80** |

# F. Discrepancy Preference Optimization Training Details

We train the LoRA model for each client following the procedure described in Section 3.3. Only the LoRA parameters are updated, ensuring efficient personalization while keeping the diffusion backbone shared across all clients.

In standard Direct Preference Optimization (DPO), it is common practice to first perform Supervised Fine-Tuning (SFT) on the pretrained model with a small amount of data, followed by preference optimization. However, both computational overhead and limited data availability constrain the effectiveness of SFT in our setting. To ensure that Diffusion fine-tuning begins with an initial discrepancy from the reference model, we introduce a non-zero initialization of the LoRA parameters (e.g., Gaussian initialization), so that the fine-tuned model and the reference model produce distinguishable outputs from the very beginning.

In our experiments, we adopt the Low-Rank Adaptation (LoRA) framework for efficient fine-tuning. Specifically, we configure LoRA with a rank of 4, and a scaling factor of 4. The adaptation is applied to the attention projection layers, including to_q, to_k, to_v, and to_out.0.

For computational efficiency and in line with our design that data supplementation is most critical in the early stages of training, we restrict DPO finetuning to only the first $T_0$ rounds. Empirically, we find that applying DPO training for merely 3-5 rounds is sufficient to yield improvements, without incurring significant additional overhead.

## G. Image Generation Details

Specifically, we use the predefined prompt set: "a photo of a {class}", "a blurry photo of a {class}", "a black and white photo of a {class}", and "a photo of a small {class}" to generate more diverse images.

We adopt the Stable Diffusion v1-4 model to synthesize images, using the default hyperparameters provided in the official implementation. The generation process is guided by a DDIM scheduler with 50 inference steps, and a guidance scale of 7.5. For each prompt, we first generate image at the default resolution of $512 \times 512$ and then resize them to match the target datasets: $32 \times 32$ for CIFAR-10 and CIFAR-100, and $64 \times 64$ for Tiny-ImageNet. The sampling temperature is kept at its default value. Through preliminary testing, we found that directly generating from $32 \times 32$ or $64 \times 64$ noise leads to poor image quality; hence, we adopt this two-step procedure to ensure both fidelity and compatibility with datasets.

## H. Scaling Analysis

### H.1. Scaling Behavior with Respect to the Number of Clients

We increase the number of clients to 30 and 50 while keeping all configurations unchanged. We find that for moderate values of $N$, the additional overhead introduced by our method remains at a reasonable level.

1. **Image Generation**: Synthetic images are shared by all clients. This stage remains constant with respect to the number of clients.

2. **Local/Global Model Inference**: The computation increases mildly with the number of clients but remains negligible compared with client local training.

3. **DPO Training**: The measured DPO execution times are listed in Table 11.

*Table 11.* DPO training overhead with varying number of clients.

| # Clients | 10 | 30 | 50 |
|---|---|---|---|
| Time (s) | 176.72 | 413.92 | 821.28 |
| Ratio (DPO / total) | 4.65% | 10.78% | 21.6% |

The experiments reported above assume comparable computational capacity for both the server and the clients. However, in real-world deployments with a large number of clients, client devices typically possess weaker compute resources than the server. Note that server-side computation and client-side local training proceed concurrently, so the server can complete most of its additional workload while waiting for clients to finish their local updates, without extending the end-to-end round duration.

### H.2. Computationally Efficient FedPDG

For extreme FL settings where the number of clients $N$ becomes very large, the per-client DPO training introduces $O(N)$ computational overhead on the server side. To address this scalability concern, we introduce *Computationally Efficient FedPDG*, a variant that trains a single shared LoRA module across all clients.

**Method.** Instead of training $N$ separate LoRA modules, this variant collects (generated sample, reward) pairs from all clients and trains one shared LoRA module. Specifically, for each client $i$, we compute the reward $r_{i,j} = \hat{p}_g(x_j) - \hat{p}_i(x_j) +$

$\lambda \cdot \text{disagree}(x_j)$ for generated samples. All samples are then pooled together and sorted by their reward values. We retain the same total number of training samples as the original per-client DPO training to match the computation budget, selecting samples with the highest rewards to form preference pairs for DPO optimization.

**Results.** Table 12 presents the complete comparison between FedPDG and its computationally efficient variant across CIFAR-10 under various heterogeneity levels.

*Table 12.* Performance comparison on CIFAR-10 between FedPDG and its computationally efficient variant (shared LoRA). We report mean accuracy (%) over three runs. **Bold** indicates the best result; underline indicates the second best.

| Method | $\alpha = 0.05$ | $\alpha = 0.1$ | $\alpha = 0.5$ | $\alpha = 1.0$ |
|---|---|---|---|---|
| FedAvg | 54.47 | 55.40 | 48.64 | 54.30 |
| FedFA | 48.22 | 52.40 | 49.24 | 52.98 |
| Gen-FedSD | 65.90 | 60.51 | 50.87 | 56.95 |
| GenFL | 56.40 | 57.27 | 49.78 | 54.90 |
| FedPDG | **67.58** | **63.82** | **54.84** | **60.14** |
| FedPDG (shared LoRA) | 66.38 | 61.65 | 53.51 | 59.24 |
| $\Delta$ (FedPDG $\rightarrow$ shared LoRA) | -1.20 | -2.17 | -1.33 | -0.90 |

This demonstrates that the computationally efficient variant provides a practical trade-off between computational cost and model performance, making FedPDG viable for large-scale federated deployments.

### H.3. Number of Trials and Statistical Significance

For each dataset and each heterogeneity setting, we perform three independent runs under identical experimental configurations. As CIFAR-10 shows some overlap in the reported mean and standard deviation values, we conduct the following statistical significance tests comparing FedPDG with the strongest baseline on CIFAR-10:

- **Paired t-test**: evaluates whether the mean improvement of FedPDG is significantly greater than zero.

- **Wilcoxon signed-rank test**: assesses whether this improvement is consistently positive without assuming normality.

- **Sign test**: verifies whether the observed win-loss pattern could occur by chance.

For all three tests, a $p$-value below 0.05 indicates that the observed improvement is unlikely to be due to random fluctuations. Table 13 summarizes the results.

*Table 13.* Statistical significance tests comparing FedPDG with the strongest baseline (Gen-FedSD) on CIFAR-10.

| Test | $p$-value | Significant ($p < 0.05$) |
|---|---|---|
| Paired t-test | $1.2 \times 10^{-4}$ | ✓ |
| Wilcoxon signed-rank test | 0.0024 | ✓ |
| Sign test | 0.019 | ✓ |

All $p$-values are well below the 0.05 threshold, confirming that, despite the visual overlap of mean and standard deviation ranges, FedPDG consistently and significantly outperforms the strongest baseline on CIFAR-10.

## I. Dataset Construction Details

The original CIFAR-10, CIFAR-100, and Tiny-ImageNet benchmarks are class-balanced in both their training and test splits. To simulate a federated setting where the *global* label distribution is itself imbalanced, we construct the client partitions and the test set as follows.

**Client partition.** Each client $i$ is assigned a fixed budget of $N$ training samples ($N = 500$ for CIFAR-10, $1000$ for CIFAR-100, $1500$ for Tiny-ImageNet). For each client, we sample a label proportion vector

$$q_i \sim \text{Dirichlet}(\alpha \cdot \mathbf{1}_C),$$

where $C$ is the number of classes and $\alpha$ controls the heterogeneity level. Client $i$ is then allocated exactly $\lfloor N \cdot q_i(c) \rfloor$ samples from class $c$, drawn without replacement from the original training pool.

**Induced global distribution.** The global label distribution is not preset, but *induced* by aggregating client allocations. Let $n_c = \sum_{i=1}^{M} \lfloor N \cdot q_i(c) \rfloor$ denote the total number of training samples from class $c$ across all $M$ clients. The global label distribution is then

$$P_{\text{global}}(y = c) = \frac{n_c}{\sum_{c'} n_{c'}}.$$

Because each $q_i$ is drawn independently from a Dirichlet prior, the aggregated counts $\{n_c\}$ are themselves imbalanced—and crucially, the degree of imbalance grows as $\alpha$ decreases.

**Test set reshaping.** The original test set is balanced across classes, which would yield a train–test distribution mismatch under our construction. To make evaluation reflect deployment, we resample the test set so that the number of test samples from each class $c$ is proportional to $n_c$. This ensures $P_{\text{test}}(y) = P_{\text{global}}(y)$.

**Distinction from CIFAR-10-LT.** This construction differs from standard long-tailed benchmarks such as CIFAR-10-LT: our test set is reshaped to match the induced global distribution, whereas CIFAR-10-LT keeps the test set balanced regardless of training imbalance. The latter choice rewards rebalancing techniques by construction; ours instead rewards methods that align with the true deployment distribution.

We show a visualization of test distribution on CIFAR-10 in Figure 10.

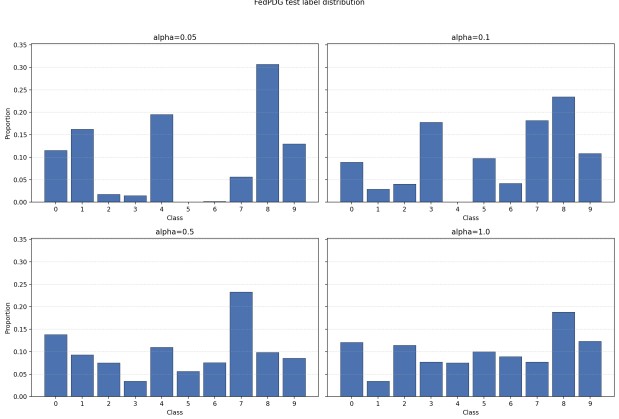

*Table 14.* Top-1 accuracy (%) on iNaturalist under different heterogeneity levels $\alpha$.

| Method | $\alpha = 0.1$ | $\alpha = 0.5$ | $\alpha = 1.0$ |
|---|---|---|---|
| FedAvg | 34.90 | 39.80 | 42.70 |
| FedProto | 35.80 | 40.90 | 43.80 |
| Gen-FedSD | 38.35 | 43.52 | 46.78 |
| **FedPDG** | **40.12** | **45.27** | **48.31** |

*Figure 10.* Our imbalanced test data distribution visualization.

*Table 15.* Hyperparameters Sensitivity experiments of $\tau$ on Tiny-ImageNet

| **Tiny-ImageNet** | $\tau = 0.001$ | $\tau = 0.003$ | $\tau = 0.01$ | $\tau = 0.013$ | $\tau = 0.016$ |
|---|---|---|---|---|---|
| FedPDG | 22.41 | 22.98 | **23.27** | 23.19 | 22.74 |

*Table 16.* Hyperparameters Sensitivity experiments of $\tau$ on CIFAR-100

| **CIFAR-100** | $\tau = 0.004$ | $\tau = 0.007$ | $\tau = 0.01$ | $\tau = 0.013$ | $\tau = 0.016$ |
|---|---|---|---|---|---|
| FedPDG | 37.14 | 37.12 | **37.41** | 37.18 | 36.95 |

## J. Experiment of Domain applicability

To validate the effectiveness of FedPDG on specific domain, we conduct an experiment on iNaturalist (Horn et al., 2018), a fine-grained species dataset whose class frequencies are skewed by local ecology. As shown in Table 14, FedPDG remains the strongest method at every heterogeneity level.

## K. Scaling to larger data and models.

The main experiments adopt a small per-client dataset and a lightweight ResNet-8 backbone for CIFAR-10, in order to simulate the practically important regime of limited data under high heterogeneity. To verify that FedPDG continues to benefit when both data and model capacity are scaled up, we conduct an additional experiment with $4,000$ training samples and $4,000$ supplemental samples per client, using ResNet-18 as the backbone. Results are reported in Table 17. FedPDG remains the strongest method across all heterogeneity levels.

*Table 17.* Accuracy (%) on CIFAR-10 with $4,000$ training and $4,000$ supplemental samples per client, using a ResNet-18 backbone. Mean $\pm$ std over three runs.

| Method | $\alpha = 0.05$ | $\alpha = 0.1$ | $\alpha = 0.5$ | $\alpha = 1.0$ |
|---|---|---|---|---|
| FedAvg | $65.93 \pm 2.68$ | $69.31 \pm 1.95$ | $75.41 \pm 0.83$ | $76.22 \pm 0.29$ |
| FedProto | $68.84 \pm 1.56$ | $72.91 \pm 1.24$ | $77.27 \pm 0.78$ | $78.44 \pm 0.40$ |
| Gen-FedSD | $73.96 \pm 1.10$ | $75.24 \pm 1.38$ | $79.11 \pm 0.74$ | $80.61 \pm 0.35$ |
| **FedPDG** | $\mathbf{74.86 \pm 1.24}$ | $\mathbf{77.12 \pm 1.08}$ | $\mathbf{81.44 \pm 0.61}$ | $\mathbf{83.20 \pm 0.34}$ |

## L. Use of LLM

Large Language Models (LLMs), specifically OpenAI's ChatGPT (GPT-4/5), were used to assist with translation and with polishing the grammar, wording, and fluency of the manuscript. All scientific ideas, experimental designs, analyses, and conclusions are solely the authors' original work. The LLM was not used to generate research content, results, or references. The authors reviewed and verified all text produced with LLM assistance to ensure accuracy and integrity.

