# OpenReview forum: "FedPDG: Prediction Discrepancy–Guided Data Generation for Heterogeneous Federated Learning"
_ICML.cc/2026/Conference — ICML 2026 regular_

### Official Review · Reviewer_wfFu · 2026-03-11

**Soundness:** 3
**Presentation:** 2
**Significance:** 3
**Originality:** 3
**Overall Recommendation:** 4
**Confidence:** 3

**Summary:**

This paper proposes FedPDG, a framework for addressing data heterogeneity (non-IID) in federated learning. Unlike existing methods that assume a balanced global class distribution, FedPDG allows the global distribution itself to be imbalanced. The core idea is to leverage the prediction discrepancy between local and global models on synthetic samples to identify deficient regions in each client's data distribution, and selectively supplement clients with synthetic data to implicitly align local distributions with the unknown global distribution. Specifically, the framework consists of three components: (1) Prediction Discrepancy-guided data Selection (PDS) with confidence-aware filtering; (2) an Exponential decay Scheduler (EXS) that controls the supply quantity of synthetic data per round; and (3) Discrepancy Preference Optimization (DPO), which personalizes the diffusion model's generative preference for each client by training lightweight LoRA modules. Experiments are conducted on CIFAR-10, CIFAR-100, and Tiny-ImageNet, where FedPDG outperforms existing methods across various heterogeneity levels.

**Compliance With Llm Reviewing Policy:**

Affirmed.

**Final Justification:**

The rebuttal has improved my assessment of the paper, and I have updated my score upward.

**Key Questions For Authors:**

Q1. Please provide detailed specifications of how the imbalanced global distribution is constructed.

Q2. Theorem 3.1 assumes all clients share the same class-conditional density q(x|y). In practical FL, different clients may exhibit covariate shift (i.e., feature distributions for the same class differ across clients). How reliable is the prediction discrepancy signal under such conditions? Is there experimental or theoretical analysis supporting the method's robustness under covariate shift? Providing such evidence would significantly strengthen the paper.

Q3. In the EuroSAT experiment (Table 6), FedPDG underperforms FedProto at $\alpha$ = 1.0 (87.62 vs. 90.50). The authors attribute this to domain shift, but could you provide a more specific analysis: under what conditions might synthetic data augmentation actually be harmful?

**Limitations:**

See weaknesses.

**Strengths And Weaknesses:**

Strengths:
1. Practical motivation: Existing data augmentation methods in FL universally assume a balanced global distribution, yet real-world global distributions are often inherently imbalanced.
2. Practical framework design with zero client-side overhead: All additional computation in FedPDG (synthetic data generation, PDS filtering, DPO training) is performed on the server side. Clients do not need to run large generative models and incur no extra computation or communication overhead.
3. Plug-and-play modular design: Experiments demonstrate that FedPDG can be combined with existing methods such as FedProx, FedFA, and FedETF (Tables 7 and 8), with further performance improvements upon combination, showing good compatibility and flexibility.

Weaknesses:
1. Strong assumptions in Theorem 3.1, with insufficient discussion of the theory-practice gap: The proof relies on two key assumptions: (a) all clients share identical class-conditional densities q(x|y), differing only in class priors; (b) models have sufficient capacity and reach Bayes optimality. Assumption (a) does not hold under covariate shift scenarios, and assumption (b) is particularly unrealistic in the early stages of FL training when models are undertrained. The paper does not adequately discuss robustness when these assumptions are violated. Some empirical experiments or theoretical analysis are needed.
2. The construction of imbalanced global distributions in experiments is insufficiently described. The paper mentions "modifying the test data distribution to approximate the global label distribution" to simulate globally imbalanced scenarios, but does not provide specific parameters (e.g., imbalance ratio, sampling strategy).
3. The confidence threshold $\tau$ = 1/C across different datasets: The paper fixes $\tau$ = 1/C (random guessing probability), but on datasets with vastly different numbers of classes (1/10 for CIFAR-10 vs. 1/100 for CIFAR-100 vs. 1/200 for Tiny-ImageNet), the meaning and filtering strength of this threshold vary significantly. Some analysis or ablation studies are needed.

---

> ### Author Rebuttal · Authors · 2026-03-31
>
> # W1
> We thank the reviewer for this important concern.
>
> For assumption (a), we analyze a setting where the clients' class-conditional distributions are not identical. Empirically, we added covariate-shift experiments on the PACS dataset, which contains 4 visual domains (art painting, cartoon, photo, and sketch). We construct a heterogeneous federated setting in which each client draws data from a domain-skewed mixture of PACS domains, while label imbalance is independently induced via a Dirichlet split. This creates simultaneous label skew and domain shift across clients. The results are below, demonstrating even when assumption (a) is violated, FedPDG still consistently outperforms baselines.
>
> | Method | 0.1 | 0.5 | 1.0 |
> |---|---|---|---|
> | FedAvg | 61.42 | 66.18 | 68.74 |
> | FedProto | 62.35 | 67.01 | 69.52 |
> | Gen-FedSD | 63.84 | 68.27 | 70.41 |
> | FedPDG | **65.96** | **69.88** | **71.93** |
>
> For assumption (b), we design the exponential scheduler to explicitly balance this issue. In the very early stage, we allocate more supplementation to quickly improve the model's representative ability. We draw an empirical inspiration that within a few training epochs, the model can acquire basic discriminative ability, which can produce an informative discrepancy signal. In the later stage, the generated data can more accurately target those classes that are underrepresented relative to the global distribution.
>
> We clarify that this theory provides intuition for our method, and we design these strategies to mitigate the theory-practical gap. Case study shows that our method works under this intuition, and achieves better alignment with the global distribution than all baselines.
> # W2
>
> For each client $i$, a label proportion vector is sampled as $q\_i \sim \text{Dirichlet}(\alpha \cdot \mathbf{1}\_C)$. Each client is then allocated exactly $N$ training samples, where the number drawn from class $c$ is exactly $\lfloor N \times q\_i(c) \rfloor$. Then, we aggregate all training labels across all clients to obtain the global label counts $\{n\_c\}\_{c=1}^{C}$, and the global label distribution is defined as $P\_{\text{global}}(y=c) = n\_c / \sum\_{c'} n\_{c'}$. The test set is then resampled by drawing from each class a number of samples proportional to $n_c$. This ensures that $P\_{\text{test}}(y) = P\_{\text{global}}$.
>
> This is the key distinction from datasets such as CIFAR-10-LT, where the test set remains balanced regardless of training imbalance. We show a visualization of test distribution here: https://anonymous.4open.science/r/F-E9A9/l.png
> # W3
>
> We thank the reviewer for this concern. We already provide a hyperparameter sensitivity analysis for $\tau$ in Section 4.5 (Figure 7). We agree that the effective filtering strength of $\tau = 1/C$ varies across datasets with different numbers of classes. To address this, we conducted additional sensitivity analyses on CIFAR-100 (C=100) and Tiny-ImageNet (C=200) by grid search. Results are shown below.
>
> **On CIFAR-100:**
>
> | τ | 0.004 | 0.007 | 0.01 | 0.013 | 0.016 |
> |---|---|---|---|---|---|
> | FedPDG | 37.14 | 37.12 | **37.41** | 37.18 | 36.95 |
>
> **On Tiny-ImageNet:**
>
> | τ | 0.001 | 0.003 | 0.005 | 0.007 | 0.09 |
> |---|---|---|---|---|---|
> | FedPDG | 22.41 | 22.98 | **23.27** | 23.19 | 22.74 |
>
> These results confirm that $\tau = 1/C$ remains a robust default across datasets.
> # Q1
>
> Please refer to W2.
>
> # Q2
>
> Please refer to W1.
>
> # Q3
>
> Thank you for this question. This phenomenon can occur in conditions when the generator's pretraining distribution is far from the target domain (e.g., natural images vs. satellite imagery or medical imaging) and heterogeneity is mild (α = 1.0).
>
> However, this limitation is mainly caused by the domain mismatch between generated data and training data, rather than by the framework itself. This can be mitigated by using domain-specific generator. As discussed in our response to Reviewer yrbB Weakness 2, domain-specific generative models are now widely available. To validate this, we conducted additional experiments on EuroSAT using a satellite-specific generator. Results are shown below:
>
> | Method | 0.05 | 0.1 | 0.5 | 1.0 |
> |---|---|---|---|---|
> | FedAvg | 51.74 | 61.92 | 82.09 | 85.94 |
> | FedProto | 58.99 | 77.19 | 90.98 | 90.50 |
> | Gen-FedSD (normal SD) | 63.29 | 78.40 | 87.14 | 85.33 |
> | FedPDG (normal SD) | 67.30 | 82.46 | 91.10 | 87.62 |
> | Gen-FedSD (SAT SD) | 68.84 | 81.63 | 89.92 | 89.91 |
> | FedPDG (SAT SD) | **70.58** | **84.61** | **92.58** | **91.68** |

---

> > ### Author Rebuttal · Reviewer_wfFu · 2026-04-01
> >
> > I thank the authors for their detailed rebuttal. I am considering changing my score from 3 to 4.

---

> > > ### Author Response · Authors · 2026-04-03
> > >
> > > Thank you for your positive acknowledgement. We appreciate your constructive suggestions on the theorem assumptions, and we will incorporate all the relevant content into the revised version.

---

### Official Review · Reviewer_yrbB · 2026-03-12

**Soundness:** 3
**Presentation:** 4
**Significance:** 1
**Originality:** 3
**Overall Recommendation:** 4
**Confidence:** 4

**Summary:**

The paper addresses label distribution skew in federated learning through a synthetic data generation approach based on diffusion models. It focuses on scenarios where the global data distribution is inherently imbalanced, a setting often overlooked in prior work that typically assumes balanced global data. The proposed method, FedPDG, supplements client datasets with selectively generated synthetic samples to better align local data distributions with the global distribution, rather than balancing local datasets. To achieve this, the framework assigns samples based on the prediction discrepancy between the global and local models, prioritizing cases where the global model performs better and predicts with high confidence, indicating underrepresented regions in the client’s data. The amount of synthetic data provided during training is controlled via an exponential decay schedule, emphasizing supplementation during the early stages of training. In addition, the pre-trained diffusion generator is personalized using direct preference optimization with an introduced reward function to produce samples that better capture client-specific distribution gaps. Experiments show that FedPDG consistently improves performance over several baselines while introducing only modest computational overhead.

**Compliance With Llm Reviewing Policy:**

Affirmed.

**Final Justification:**

My main concern has been largely addressed by the rebuttal, so I am increasing my score. However, I ask the authors to include MCC, balanced accuracy, and macro-F1 across the full set of experimental results in the camera-ready version, along with the additional details I requested about the dataset construction, and a clearer discussion of when preserving the global imbalance is appropriate and when it may not be desirable.

**Key Questions For Authors:**

1. Why is it desirable to match the global class imbalance rather than mitigate it during training?
2. What is a practical use case where a diffusion model can capture the domain and a biased model would be desirable?
3. How does the method ensure that minority classes are predicted reliably, especially under severe imbalance?
4. Could the authors report metrics suited for imbalanced datasets (e.g., MCC, balanced accuracy)?
5. Could the paper include centralized baselines (including centralized training with class-balancing techniques)?

**Limitations:**

The authors acknowledge the limitation that different domains in real-world applications may pose challenges for a pre-trained, general-purpose generator. To address this, they demonstrate results on satellite imagery. However, the current evaluation still focuses primarily on benchmark datasets with relatively simple visual domains. Evaluating the approach on more challenging or domain-specific datasets would strengthen the empirical validation. In addition, providing more details about which domains can still be effectively captured by the model and which cannot would further improve the paper.

**Strengths And Weaknesses:**

Strengths:
+ Addressing FL under globally imbalanced data distributions is important and largely underexplored.
+ The method introduces no additional client-side computation and only moderate server overhead, mainly in early rounds.
+ The components of the approach are intuitive and appear to contribute meaningfully to performance.
+ The paper proposes an approximate variant to support scenarios with many clients and thus addresses scalability
+ The paper is well written and all methods and experiments are understandable.

Weaknesses:
- The overall accuracies on CIFAR-10, which are around 60%, seem low to me. I assume this might be because fewer samples are used in total due to the global data imbalance. However, this is not entirely clear and should be explained in the paper. There are too few details about how the global dataset is imbalanced. Since this is one of the main contributions of the paper, please explain precisely how the originally balanced benchmark dataset is made imbalanced and how it is subsequently distributed.
- The benchmark datasets are relatively simple tasks, and diffusion models are typically pre-trained on similar data and classes. What would be a practical real-world use case where we intentionally want a bias in the model and where the domain can still be captured by a pre-trained diffusion model? I assume one limitation is that some domains cannot be represented. For example, a diffusion model would likely not be able to generate CT images. More generally, what are the limits of the pre-trained diffusion model used here?
- Minor comment: Please indicate the dataset used in the captions of Table 7 and Table 8.

**Major Concern:**

A central premise of the paper is that aligning local datasets with the globally imbalanced distribution is beneficial for training the global model. I am convinced that the proposed method is appropriate for this goal and the empirical results suggest it does this better than the baselines. However, I am not convinced that this objective itself is desirable.
In many centralized learning settings, class imbalance is typically mitigated rather than preserved (e.g., via augmentation, reweighting, or resampling) to prevent models from becoming biased toward majority classes and neglecting minority classes. Matching the global imbalance may therefore risk reinforcing dataset bias, whereas the goal is often to train models that perform well across all classes, including minority ones.
The current evaluation does not fully resolve this concern because the paper primarily reports accuracy, which can be misleading for imbalanced datasets. For example, a model that predicts the majority class most of the time may achieve high accuracy without meaningfully learning minority classes. Without metrics that account for true/false positives and negatives (e.g., MCC, balanced accuracy, macro-F1), it is difficult to determine whether the method improves classification performance across classes or mainly improves alignment with the global bias.
To better support the paper’s core claim, I would suggest the following additional comparisons:
- Report metrics suited for imbalanced classification (e.g., MCC, balanced accuracy, macro-F1).
- Include centralized baselines on the same imbalanced datasets.
- Include a centralized baseline that explicitly mitigates imbalance (e.g., via augmentation or reweighting).
- In the ablation study, evaluate a variant of the proposed method where synthetic samples aim to balance local datasets instead of aligning them with the global imbalance.

**I am happy to increase my score if this concern can be addressed convincingly.**

---

> ### Author Rebuttal · Authors · 2026-03-31
>
> # Main Concern
> Thank you for this thoughtful concern. We believe this is a fundamental question about when class imbalance should be preserved versus mitigated, and we appreciate the opportunity to clarify our reasoning.
>
> In many applications, like cancer diagnosis, rebalancing is necessary.  This is because the cost of misclassifying a positive case (false negative) vastly outweighs a false alarm (false positive), so the model need to be deliberately biased toward the minority class (cancer-positive), even at the expense of overall accuracy.
>
> However, there are many real-world deployments operate under the “equal-cost assumption”, where preserving the natural class distribution is beneficial:
>
> 1. E-commerce product classification. In federated product classification across merchants, some categories (e.g., phone cases) naturally have far more items than others. Misclassifying a phone case as a charger is no more or less harmful than misclassifying a vinyl record as a CD. The system's objective is to maximize overall classification correctness, which is best served by training on the true product distribution.
>
> 2. Wildlife and plant species recognition. A federated system aggregating images from camera traps or nature-identification apps will naturally have far more photos of common species (e.g., deer, sparrows) than rare ones. For applications such as ecological surveys where the goal is accurate species identification across all submitted photos, the model should reflect the true species frequency distribution rather than artificially equalizing rare and common classes.
>
> 3. Web image content classification. In federated content classification across heterogeneous web platforms, category frequencies are naturally skewed and reflect the true composition of web data. When misclassification costs are roughly symmetric, preserving this natural distribution is more appropriate.
>
> Regarding the concern that accuracy improvements may due to majority-class bias rather than genuine learning, we conducted additional experiments on cifar-10 dataset setting in our paper with a single seed. Following your advice, we compare our method with several baselines, and report (Test acc / Balanced acc / Macro-F1 / MCC) below.
>
> | Method | α=0.1 | α=0.5 | α=1.0 |
> |---|---|---|---|
> | FedAvg | 47.324 / 45.646 / 45.067 / 0.395 | 50.692 / 48.963 / 49.044 / 0.444 | 53.786 / 51.790 / 51.352 / 0.481 |
> | FedPDG (balanced) | 52.916 / 51.281 / 50.757 / 0.462 | 53.818 / 51.502 / 51.889 / 0.478 | 55.889 / 52.392 / 52.717 / 0.503 |
> | FedPDG | **55.081 / 53.944 / 53.033 / 0.488** | **55.201 / 55.510 / 54.862 / 0.496** | **56.671 / 55.460 / 54.308 / 0.517** |
> | Centralized | 57.847 / 56.205 / 55.992 / 0.517 | 54.059 / 52.501 / 52.421 / 0.482 | 57.512 / 54.642 / 54.588 / 0.522 |
> | CentralizedAugment (towards balanced) | 60.313 / 59.513 / 58.926 / 0.547 | 58.208 / 56.962 / 57.173 / 0.528 | 60.276 / 57.634 / 57.584 / 0.553 |
> | CentralizedAugment (towards true imbalanced) | **61.335 / 59.607 / 59.708 / 0.557** | **60.072 / 59.951 / 59.717 / 0.550** | **61.478 / 59.671 / 59.170 / 0.568** |
>
> The results show that our method improves performance across all classes, not merely the frequent ones.
>
> # W1
> For the overall accuracy, please refer to our reply to Reviewer 8a3B W1. For the dataset construction, please refer to our reply to Reviewer wfFu W2.
> # W2
> Thank you for this question. Regarding the limits of pre-trained diffusion models, various domain-specific diffusion models are increasingly available. For example, CT[1], X-ray[2], satellite [3] and wild animal[4].
>
> Based on these works, a complex task is wild animal species classification, where species distributions are imbalanced caused by local ecology. To validate our methods in this domain, we conducted additional experiments on the iNaturalist dataset[5] using model[6].
> | Method | 0.1 | 0.5 | 1.0 |
> |---|---|---|---|
> | FedAvg | 34.90 | 39.80 | 42.70 |
> | FedProto | 35.80 | 40.90 | 43.80 |
> | GenFedSD | 38.35 | 43.52 | 46.78 |
> | FedPDG | **40.12** | **45.27** | **48.31** |
>
> Regarding domain coverage, we claim that current various generative models cover the vast majority of real-world scenarios. The limitations are narrow: domains requiring highly specialized domain knowledge (e.g., molecular structures, circuit schematics) or extremely rare disease with no publicly available training data for generator fine-tuning. We will add an explicit domain applicability discussion to the revised manuscript.
>
> [1] GenerateCT: Text-Conditional Generation of 3D Chest CT Volumes
>
> [2] SV-DRR: High-Fidelity Novel View X-Ray Synthesis Using Diffusion Model
>
> [3] DiffusionSat: A Generative Foundation Model for Satellite Imagery
>
> [4] TaxaDiffusion: Progressively Trained Diffusion Model for Fine-Grained Species Generation
>
> [5] The iNaturalist Species Classification and Detection Dataset
>
> [6] Qwen-Image Technical Report
> # Q1 & Q3 & Q4 & Q5
> Please refer to our reply to your major concern.
> # Q2
> Please refer to W2.

---

> > ### Author Rebuttal · Reviewer_yrbB · 2026-04-03
> >
> > Thank you for the rebuttal and the additional experiments.
> >
> > The new results make the argument for learning the imbalanced global distribution more convincing. However, I still believe that the motivation is highly context-dependent and may not be desirable in some domains (such as medicine), as the authors also acknowledge in the rebuttal.
> >
> > Since my main concern has been largely addressed, I am willing to increase my score. For the revised manuscript, please include MCC, balanced accuracy, and macro-F1 across the full set of experimental results, along with the additional details I requested about the dataset construction. It would also be valuable to include a clearer discussion of when preserving the global imbalance is appropriate, and when it may not be desirable.

---

> > > ### Author Response · Authors · 2026-04-03
> > >
> > > Thank you for your positive acknowledgement and for your willingness to raise your score. We will include MCC, balanced accuracy, and macro-F1 across the full set of experimental results in the revised manuscript, along with detailed dataset construction information and a clearer discussion of when preserving global imbalance is appropriate versus undesirable.

---

### Official Review · Reviewer_aBeh · 2026-03-12

**Soundness:** 3
**Presentation:** 3
**Significance:** 3
**Originality:** 3
**Overall Recommendation:** 5
**Confidence:** 4

**Summary:**

This paper proposes a prediction-discrepancy–guided data generation framework for heterogeneous federated learning under unknown global label distributions. The key idea is to leverage the discrepancy between the losses of the global model and client models on  generated samples to infer which classes are underrepresented on each client. The method further adapt the diffusion generator via DPO paradigm so that it can generate samples that induce large prediction discrepancies. Experiments on CIFAR-10/100, Tiny-ImageNet, and EuroSAT demonstrate improved performance over prior federated learning baselines and generative data augmentation methods.

**Compliance With Llm Reviewing Policy:**

Affirmed.

**Final Justification:**

The rebuttal addresses my concerns. I would keep my score.

**Key Questions For Authors:**

1.Does the algorithm include any validation steps to prevent the diffusion models from generating low-quality images?

2.For the scheduler part, I notice it also influences a lot. Though there is empirical results in the paper, can you give an explanation why exponential decay scheduler perform best?

**Limitations:**

Yes. The paper discusses some limitations in domain heterogeneity scenario, in Section 4.3 "domain shift of the diffusion model".

**Strengths And Weaknesses:**

### **Strengths**

1.The paper addresses a realistic and underexplored setting in federated learning: unknown global class imbalance with client heterogeneity. The use of prediction discrepancy signal is reasonable with theoretical foundation under a label-shift assumption, and the algorithm can be easily implemented.

2.The paper is well structured and the method description is clear.

3.The proposed discrepancy-guided mechanism provides a novel way to adaptively allocate synthetic data without requiring access to true global label information. The method is potentially useful for practical FL deployments where each client cannot reliably estimate global data distributions due to privacy constraints.

4.The paper introduces a discrepancy-based signal for guiding synthetic data supplement, which is distinct from prior class-balancing methods. The DPO training with discrepancy reward between global and local is also a creative idea for federated learning.

------

### **Weaknesses**

1.To simulate globally imbalanced scenarios, the authors manually modified the CIFAR-10 distribution. While this is acceptable, a better and more standardized choice would be to evaluate the framework on well-established long-tailed benchmarks, such as CIFAR-10-LT (e.g., https://huggingface.co/datasets/tomas-gajarsky/cifar10-lt). Using a standard dataset would make the empirical results more convincing and facilitate fairer comparisons for future work.

2.The Stable Diffusion model used in the experiments is relatively outdated. To my knowledge, there are many new image generative models currently available, and the paper lacks performance evaluations on these latest models.

---

> ### Author Rebuttal · Authors · 2026-03-31
>
> # W1
> We thank the reviewer for this suggestion. We agree that evaluating on well-established benchmarks strengthens the empirical contribution. However, we note a key distinction: standard long-tailed benchmarks such as CIFAR-10-LT pair an imbalanced training set with a balanced test set, targeting the setting where the goal is to recover uniform performance across all classes. Our work instead addresses a different and practically motivated setting where the real-world test distribution itself is imbalanced. Adopting a balanced test set would introduce a train-test distribution mismatch that contradicts our core problem formulation.
> To address your concern, we conducted additional experiments using different CIFAR-10-LT imbalance ratios(IF) to control imbalance degree with $\alpha=0.1$, then we resample the test set to match the same long-tailed global distribution. Results are shown below:
> | Method | IF = 10 | IF = 50 | IF = 100 |
> |---|---|---|---|
> | FedAvg | 61.20 | 57.95 | 55.34 |
> | FedProx | 61.88 | 57.41 | 57.90 |
> | FedProto | 62.14 | 59.73 | 56.21 |
> | Gen-FedSD | 61.92 | 58.02 | 56.88 |
> | FedPDG | **65.07** | **62.54** | **60.36** |
> # W2
> Thank you for this suggestion. We already included our experiments with Flux-Dev in Appendix (Table 10), which demonstrates that our method generalizes beyond SD v1-4. To further address this concern, we have conducted additional experiments with more recent generative models on origin cifar-10 0.1 settings in our paper, including Qwen-Image and Bagel. Results are shown below.
> | Method | Flux-Dev | Qwen-Image | Bagel |
> |---|---|---|---|
> | Gen-FedSD | 58.47 ± 1.38 | 60.31 ± 1.15 | 60.89 ± 1.06 |
> | FedPDG | **60.02 ± 1.17** | **62.24 ± 0.94** | **62.95 ± 0.87** |
>
> These results confirm that FedPDG is generator-agnostic: the prediction discrepancy selection and DPO fine-tuning components impose no assumptions on the underlying generative model, and performance consistently better than balance-oriented methods.
>
> # Q1
> Our framework does not include an explicit image quality validation step. Modern generative models' generation quality is stable and low-quality outputs are rare in practice. Moreover, our confidence-aware filter (Eq. 6) provides an implicit quality gate. As for DPO fine-tuning, the KL divergence term in the DPO objective penalizes large deviations from the reference model, preventing the LoRA-adapted generator from drifting into degenerate output modes.
> # Q2
>
> The exponential decay scheduler performs best because it naturally matches the two-phase dynamics of federated training.
>
> In early rounds, the global model is poorly initialized and the prediction discrepancy signal cannot yet reliably identify which classes are underrepresented for each client. A large initial supply of synthetic data at this stage compensates for this unreliability by broadly covering the data space, preventing clients from converging to poor local minima that are difficult to escape later.
>
> In later rounds, the global model has stabilized and the discrepancy signal becomes increasingly accurate. At this point, the marginal benefit of additional synthetic data diminishes — the model has already learned the global distribution structure — and reducing supply avoids introducing noise that could destabilize convergence.
>
> The exponential schedule, $N_t \propto t^{-\beta}$, provides a steep initial drop followed by a long tail, which precisely captures this asymmetry: aggressive early supplementation and gradual late-stage reduction. Linear and stepwise schedules approximate this behavior but less smoothly (in Appendix Figure 8), which explains their slightly inferior performance.

---

> > ### Author Rebuttal · Reviewer_aBeh · 2026-04-03
> >
> > Thank you for the rebuttal, which addresses my concerns. I would keep my score.

---

> > > ### Author Response · Authors · 2026-04-03
> > >
> > > Thank you for your positive acknowledgment. We are glad that our responses have addressed your concerns. We sincerely appreciate your careful evaluation of our work.

---

### Official Review · Reviewer_8a3B · 2026-03-12

**Soundness:** 3
**Presentation:** 4
**Significance:** 3
**Originality:** 3
**Overall Recommendation:** 5
**Confidence:** 4

**Summary:**

This paper proposes FedPDG, a novel framework using synthetic data to mitigate data heterogeneity in Federated Learning. Unlike prior work assuming a balanced global class distribution, FedPDG addresses scenarios where the true global distribution is unknown. The core idea of this method is to use prediction discrepancy as a signal to identify specific data regions that a client lacks relative to the global distribution.

**Compliance With Llm Reviewing Policy:**

Affirmed.

**Final Justification:**

The author's reply fully resolved my question, therefore my score remains accept.

**Key Questions For Authors:**

1. I am curious if DPO fine-tuning address domain gaps in diffusion models? If domain-specific generators are unavailable, I'm wondering if the DPO mechanism itself can bridge the domain gap.

2. Shared LoRA Strategy: For the "Computationally Efficient" variant, how do you select synthetic data for the shared LoRA training to ensure the shared module doesn't become biased toward a few dominant clients?

**Limitations:**

Yes. The authors adequately discuss limitations regarding domain shift and the computational scaling of per-client DPO training.

**Strengths And Weaknesses:**

Strengths:

- This paper aims to mitigate data heterogeneity and imbalanced global distribution, which is an important problem overlooked in current Federated Learning. The paper provides a novel signal (loss discrepancy) as a valid indicator of label scarcity. This signal is validated by a solid theoretical foundation in Theorem 3.1. The integration of DPO for generative preference in FL is also well-reasoned.

- The paper conducted extensive experiments to demonstrate the effectiveness of their method. The quantitative and qualitative case study part is easy to follow and convincing.  Table 4 shows their method aligns the client distribution towards the unknown global distribution, which validates the core insight of the approach.

- The paper provides a thorough computational overhead analysis for both the client and server sides, which is highly important for data-generative based methods in federated learning, proving that the framework is feasible for real-world deployment.

Weaknesses:

- The accuracy reported in Table 2 and Table 3 are lower than those typically seen in related federated learning papers. A reason, I guess, is that the paper uses severely small local data sizes (only 500 samples per client for CIFAR-10). While the relative improvements over baselines are clear, showing results for larger local datasets would make the effectiveness of the method more convincing.

- The Computationally Efficient Variant of FedPDG is highly useful for scaling this method in real-world scenarios. However, the description for this variant is too concise, which makes it difficult to fully grasp the exact process. The paper should explain in detail how generated samples and their corresponding rewards from clients are processed to train the single shared LoRA module.

- Minor Typos: There are some typo errors in the paper. In Section 4.5, the proposed method is incorrectly mentioned as "FedDPD" instead of "FedPDG". A thorough proofread is recommended for the final version.

---

> ### Author Rebuttal · Authors · 2026-03-31
>
> # W1
> Thank you for raising this point. We intend to simulate settings where data is limited with high heterogeneity, so we use smaller local dataset size. In addition, we adopt ResNet-8 as the backbone for cifar-10, which is a relatively small model. To verify that our method scales with more data, we conducted additional experiments with 4,000 training samples and 4,000 supplemental samples per client, with resnet18 model. Results are shown below.
> | Method | α=0.05 | α=0.1 | α=0.5 | α=1.0 |
> |---|---|---|---|---|
> | FedAvg | 65.925 ± 2.684 | 69.312 ± 1.947 | 75.408 ± 0.826 | 76.221 ± 0.291 |
> | FedProto | 68.842 ± 1.563 | 72.914 ± 1.241 | 77.268 ± 0.781 | 78.436 ± 0.402 |
> | Gen-FedSD | 73.956 ± 1.102 | 75.244 ± 1.376 | 79.105 ± 0.744 | 80.612 ± 0.351 |
> | FedPDG | **74.864 ± 1.236** | **77.118 ± 1.084** | **81.437 ± 0.612** | **83.204 ± 0.338** |
> # W2
> Thank you for this suggestion. In the efficient variant, we train a single shared LoRA using pooled data from all clients. Concretely, for each generated sample $(x\_j, y\_j)$, we compute its reward with respect to every client $i$: $r\_{i,j} = \hat{p}\_g(x_j) - \hat{p}\_i(x\_j) + \lambda \cdot \text{disagree}(x\_j)$. All triples $(x\_{i,j}, y\_{i,j}, r\_{i,j})$ across all clients are then pooled into a single collection $R = \{(x\_j, y\_j, r\_{i,j})\}\_{i,j}$, sorted in descending order of reward. We retain the fixed number of samples to train a shared LoRA, reducing DPO overhead from $O(N)$ to $O(1)$ with respect to the number of clients. We will revise this description in the revised manuscript.
> # W3
> We will correct these typos in the revised manuscript.
> # Q1
> Thank you for this inspiring question. While our DPO fine-tuning is not explicitly designed to bridge domain gaps, we believe it can mitigate partial gap when the domain shift is moderate. Since pre-trained diffusion models already encode a rich diversity of visual styles, a proper reward signal can steer generation toward stylistically closer outputs — for instance, guiding SD to produce images with lower saturation, higher spatial frequency, or more structured patterns that better approximate the target domain. In this sense, DPO acts as a soft domain adapter when the gap is not severe.
> For large domain gaps (e.g., medical images), DPO over a small number of rounds cannot compensate for the fundamental capability boundary of the base model. In such cases, replacing SD with a domain-specific generator (see our reply to Reviewer yrbB W2) and applying our DPO fine-tuning can further mitigate the gap. We will add this discussion and more visual examples in the revised manuscript.
> # Q2
> We clarify that the shared LoRA variant is explicitly a compromise solution for scenarios where $N$ is very large. Under this constraint, using top-K samples by reward can prioritize the regions of the data space that are most underrepresented relative to the global model, which can mitigate heterogeneity between clients. Notably, in this setting, the number of clients is large and the top-K samples are drawn from a diverse pool, so that the dominance effect of any single client diminishes as $N$ grows, making the shared LoRA a practically effective approximation.

---

> > ### Author Rebuttal · Reviewer_8a3B · 2026-04-03
> >
> > I appreciated the authors for providing comprehensive and insightful responses. My concerns have been addressed with reasonable explanations. It is a solid work. I would keep my positive score.

---

> > > ### Author Response · Authors · 2026-04-03
> > >
> > > Thank you for your positive acknowledgement. We are glad our rebuttal addressed your concerns. We will incorporate the relevant content into the revised version.

---

### Decision · Program_Chairs · 2026-04-30

**Decision:**

Accept (regular)

**Comment:**

The paper introduces a method that leverages a pre-trained a diffusion model to generate synthetic samples and use the prediction discrepancy (in terms of the loss values) to estimate the local client distributions. Without assuming a balanced global distribution, it assumes the test distribution is the same as the training client data's distribution. The use of loss discrepancy is intuitive and lightweight as a metric of distribution gaps. The proposed method mostly happens on the server side, making the method feasible for resource-constrained settings. The authors conducted a variety of experiments to demonstrate the utilities of the proposed method. All reviewers lean towards acceptance, and I courage the authors to incorporate all feedback into the revision.